# A comprehensive survey and analysis of international drinking water regulations for inorganic chemicals with comparisons to the World Health Organization's drinking-water guidelines

**Erika J. Mitchell[1], Seth H. Frisbie** [2]*

**1** Better Life Laboratories, Inc., Calais, VT, United States of America, **2** Department of Chemistry and Biochemistry, Norwich University, Northfield, VT, United States of America

* sfrisbie@norwich.edu

**Data Availability Statement:** All relevant data are within the paper and its Supporting Information files.

## Abstract

### Background

The World Health Organization (WHO) has published criteria for determining the quality of drinking water since 1958. Since 1984, these criteria were termed "guidelines" to emphasize that they are not national standards, but rather guidelines for nations to develop their own national standards, which may take into account local environmental, social, economic, and cultural conditions. When calculating guideline values (GVs), the WHO reviews the toxicological literature, calculates a health-based value (HBV), and determines whether the HBV should be adopted as a GV. The WHO also considers aesthetic aspects of drinking water quality, such as taste and the staining of plumbing fixtures, and additionally supplies aesthetic values (AVs) for certain drinking water contaminants. There is no central registry for national drinking water standards, so the degree of variation of national drinking water standards is not known.

### Methods

We examined standards, guidelines, and background documents for all inorganic contaminants published by the WHO from 1958–2022. We also searched for national drinking water standards for all independent countries.

### Results

We found the WHO currently has 16 GVs, six HBVs without GVs, and six AVs without HBVs or GVs for inorganic drinking water contaminants, excluding disinfection agents and their byproducts. More than half of the point of departure studies used to support these values were published in 2005 or earlier. Ninety-eight percent of the world's population lives in jurisdictions with drinking water standards, and 14 countries directly link their national standards to the current WHO's drinking water guidelines. Lack of transparency (standards available

**Funding:** SHF received support for this project from the Norwich University Board of Fellows Faculty Development Prize. There is no grant number for this support. The Board of Fellows had no role in study design, data collection and analysis, decision to publish, or preparation of the manuscript.

**Competing interests:** The authors have declared that no competing interests exist. EJM's affiliation is with Better Life Laboratories, a nonprofit organization that conducts scientific research and provides technical expertise, equipment, and training to help needy people around the world. Better Life Laboratories received no specific funding for this project from any donors. Donors to Better Life Laboratories provided no input in choosing the subject matter of this project, the hypotheses that were tested, the method of analysis, the research findings, or the manner of disseminating the results. This does not alter our adherence to PLOS ONE policies on sharing data and materials.

only through purchase) and typographical errors are common problems, especially for resource-limited countries.

## Conclusions

The WHO drinking water guidelines are crucially important for drinking water safety; they are used for guidance or as official standards throughout the world. It is crucial that they be based on the best available science.

## Introduction

Drinking water represents a primary route of exposure to both biological pathogens and chemical contaminants. In order to protect public health by limiting exposure to such disease-causing substances, the World Health Organization (WHO) and most national governments have established drinking water guidelines or standards that are used as criteria for deciding whether water is safe for drinking.

The WHO's drinking water guidelines and many of the drinking water standards created by individual countries include specifications for biological pathogens and chemicals. Biological pathogens present in water typically are sourced through contamination by animal or human fecal matter. Chemical contaminants may be naturally occurring or sourced anthropogenically. Naturally occurring chemicals in drinking water are generally inorganic compounds, especially the 76 elements make up the earth's crust, while anthropogenic chemical water contaminants may be either inorganic or organic [1]. Some chemicals are added purposively to water for disinfection; these chemicals and their byproducts may be present in water as chemical contaminants.

The focus of this study is the chemicals that may be found in drinking water due to naturally occurring processes; thus, this study includes inorganic compounds such as dissolved minerals from the earth's crust that have been reported in drinking water. Although these inorganic compounds have been found in drinking water due to natural processes, they may also be sourced anthropogenically, but the regulations for these chemicals do not distinguish between natural or anthropogenic sources of contamination. This study does not include disinfectants and disinfectant byproducts, or organic chemical contaminants in drinking water, which are typically sourced anthropogenically.

National standards for inorganic chemical contaminants vary widely and often differ from the WHO drinking water guidelines. Not all countries have standards for every chemical contaminant for which the WHO provides guidelines, while some countries have standards for inorganic chemical contaminants for which the WHO does not provide guidelines. Some resource-limited countries have formally adopted the WHO guidelines as their national standards, while other resource-limited countries have not established any legal drinking water standards [2, 3]. The full extent of variation of national drinking water standards, the range and mode for each regulated inorganic chemical contaminant and how this may relate to national per capita income is not known since the WHO does not routinely collate all national standards.

As part of its process for setting drinking water guidelines, the WHO typically performs a risk assessment for each contaminant and calculates a health-based value (HBV) [4]. The HBV is set at a level that should protect the most sensitive population against adverse health effects when the contaminant is consumed in drinking water over a lifetime of exposure [4, 5]. That

is, the HBVs are designed to ensure that exposures to contaminants remain below tolerable daily intakes (TDIs), assuming that a 60-kilogram adult consumes two liters of drinking water per day for most contaminants, and that drinking water represents a fraction of total daily exposure [4, 5].

The WHO uses its HBVs when setting its formal guideline values (GVs) [4]. For the majority of contaminants, the GV is set to equal to the HBV. However, in some notable instances such as arsenic (As), the WHO determined that the HBV would be too difficult to achieve due to the practical concerns of "analytical achievability" and "treatment performance" and set the GV at a level that can be readily measured in a routine testing laboratory or readily removed with a conventional treatment system, but less protective of health [2, 4–6].

For some other contaminants such as iron (Fe) that may degrade the acceptability of the water for consumers by imparting an unpleasant taste or by staining laundry or plumbing fixtures, the WHO determined that a formal GV was unnecessary if the assumed acceptability threshold or aesthetic value (AV) was lower than the HBV and so declined to set a GV (WHO Fe) (lower values are more protective since they imply lower exposures).

Other reasons for not setting a GV after establishing an HBV for a contaminant include the use of the contaminant for water treatment, such as for aluminum (Al), or limited occurrence in drinking water, such as for beryllium (Be) [5, 6]. The distinction between HBVs and GVs is not widely known, so national regulators may not be aware that the list of WHO GVs may not constitute the entire list of contaminants they may need to consider when setting national standards.

A 2015 study by the WHO sought to determine the extent to which their guidelines are used and reflected in national drinking water standards of the member states of the United Nations [3]. National guidelines were obtained from members of the WHO's International Network of Drinking Water Regulators (RegNet), as well as internet searches and purchases from national standards agencies [3]. Although this was the most comprehensive survey of national drinking water standards at the time, regulations were obtained for only 104 countries and territories [3]. In a study of international arsenic (As) drinking water standards, our team searched for drinking water standards from all 193 member states of the United Nations, plus two additional states for which the World Bank provides GDP data; we found regulations for 176 out of these 195 countries [2]. In that study, we found a strong link between national income as measured by GDP per capita and the level of protectiveness of a country's As drinking water standard. We also found that 32% of the world's population live in countries where the national drinking water standard for arsenic is less protective than the WHO's GV for As of 10 µg/L [2].

These findings for national As standards suggested a need for additional analyses of national drinking water regulations to determine how well the world's population is protected from inorganic chemicals in drinking water through national regulations, and to examine how closely drinking water regulations are linked to gross domestic product (GDP) per capita. Our hypotheses were that a significant proportion of the world's population lives in jurisdictions with drinking water standards that do not protect to the levels specified in the WHO GVs or HBVs, and that resource-limited countries are less likely to have standards that protect to the levels of the WHO GVs and HBVs than countries with more economic resources.

The overall goal of this study was to document the diversity of international regulations for inorganic contaminants in drinking water. We sought to determine a complete list of the countries that have published drinking water guidelines for inorganic contaminants and analyze whether national income as measured by per capita gross domestic product is correlated with drinking water regulations, their content or availability. We also sought to determine an exhaustive list of all inorganic chemicals for which there are national drinking water

regulations, as well as to determine which chemical contaminants are most commonly regulated by national governments, and which appear in the national regulations of only a few countries. An additional goal was to compare the national regulations to the WHO drinking water guidelines, to determine the extent to which the WHO drinking water guidelines are influential for the establishment of national drinking water standard values.

## Materials and methods

### Database of World Health Organization drinking water standards and guidelines

We collated all published WHO drinking water standards [7–9] and drinking water guidelines [10–18]. For each edition and addendum of the WHO standards and guidelines, we extracted the complete list of standard values (SVs) or GVs for each inorganic contaminant. We also examined background documents for each inorganic chemical to identify the calculated HBV or stated AV and the point of departure (POD) studies on which the values were based.

### Database of national drinking water standards

We examined the legal documents in our database of national drinking water standards for arsenic and made a complete list of national drinking water standards for all inorganic chemicals [2]. As described in detail in our study of drinking water regulations for As [2], we began our original database by identifying a list of nation states including the 193 member states of the United Nations and added to this list two additional states for which the World Bank provides income data (Kosovo and Taiwan), for a total of 195 countries [19, 20]. We then searched the Food and Agriculture Organization of the United Nations FAOLEX database of international laws [21], Google [22], and Google Scholar [23] for national drinking water regulations using the search terms "drinking water quality standards", "drinking water standards", "drinking water", "water", "arsenic", "μg/L", and "mg/L". Searches were first conducted in English, then if no results were found, were continued in official national languages for each country. If we could not find any official government publications with drinking water regulations for a country, we searched Google Scholar [23] and PubMed [24] for secondary evidence of regulations in the form of published journal articles, dissertations, or theses that reference national regulations. We also determined the national organizations responsible for publishing drinking water regulations, located contact information for these organizations, and sent requests via email and/or Facebook Messenger in a national language requesting assistance with finding the national drinking water regulations.

### World Bank population, income data, and population-weighted gross domestic product/capita calculations

We used population and GDP data for the year 2020 published by the World Bank [25, 26]. When comparing the effects of various regulatory factors on the well-being of the total world population, we considered countries both as individual entities and also as proportions of world population. For many comparisons, we created population-weighted GDPs by summing the GDPs of all countries meeting a given criterium and dividing this by the sum of all populations living in these countries. We termed this value the "population-weighted GDP (P-W GDP)/capita". When calculating P-W GDP/capita values, we included all countries for which the WB provides data; when either the income or population data were missing from the WB datasets, we omitted the countries from the P-W GDP/capita calculations [25, 26]. The

countries with missing WB data were Andorra, Eritrea, Lichtenstein, Marshall Islands, Monaco, North Korea, San Marino, South Sudan, Syria, Taiwan, Turkmenistan, Venezuela, and Yemen.

## Statistics and visualizations

We performed all statistical calculations in R version R-4.1.3 "One Push-Up" [27]. The datafile and R codes created for this project are available in S1–S4 Files. This study examined a complete and finite population (countries of the world) without sampling. Thus, the study produced population parameters, not estimates based on sampling. Since the data is the entire set of national regulations and did not involve sampling, hypothesis testing was not performed. The calculated means are, by definition, population means, not sample means. Thus, differences between means represent distinct differences rather than differences that may or may not be statistically significant [28]. We created all figures and maps in R using the R packages ggplot2 and ggmaps [29, 30].

## Results and discussion

### World Health Organization standards and guidelines

A 1953 survey of Member States by the WHO identified an urgent need for drinking water quality criteria; in response to this need, in 1958 the WHO published its first standards for drinking water quality [7]. Revisions to the WHO standards were released with new editions in 1963 and 1971 [8, 9]. In 1984, the term "standard" was replaced with "guideline" to emphasize that "the levels recommended in the guidelines for water constituents and contaminants are not standards in themselves" [10]. That is, the WHO guidelines are "intended for use by countries as a basis for the development of standards", while taking into consideration the "prevailing environmental, social, economic, and cultural conditions [10]." Subsequent revisions to the WHO guidelines were released as new editions or addenda in 1993, 1998, 2004, 2006, 2008, 2011, 2017, and 2022 [11–18].

From the very beginning, the WHO has usually used risk assessments to calculate values for deciding if the concentration of a contaminant in water is safe to drink. In the 1958, 1964, and 1971 editions of the WHO standards, these values were listed as "standards", which we shall term standard values (SVs) in the ensuing discussion [7–9]. In 1984, when the WHO began using the terms "guidelines" and "guideline values" (GVs), the term "health-based values" (HBVs) also began to appear [10]. An HBV is calculated based on available scientific literature and set to a level at which no significant adverse effects to health are expected over a lifetime of exposure. In most cases, the GV is set to be equal to the HBV. However, the GV may be set to a level higher than the HBV (that is, set to a less protective level) due to practical issues such as "analytical achievability" (the lowest concentration that can be reliably measured in a routine testing laboratory) and "treatment performance" (the lowest concentration that can be reliably removed with a conventional treatment system) [18, 31]. In the 2021 revised background document for silver (Ag), the WHO introduced the term reference value (RV) for a "bounding value" that is a provisional health-based value [32]; the term Reference Value, which we shall abbreviate "RV", is used with respect to Ag in the 2nd addendum to the WHO's 4th edition of the drinking water guidelines [18].

The WHO has also noted that for some contaminants, there may be consumer acceptability thresholds above which consumers may notice unpleasant odors or tastes, or the staining of laundry or plumbing fixtures [7–18]. We shall designate these acceptability thresholds as "aesthetic values" (AVs) in the ensuing discussion. In some cases where the AV is lower than the

HBV, the WHO has decided that a formal GV is unnecessary, relying on consumer acceptability to protect health [33].

The current (2022) WHO GVs, HBVs, health-based RVs, and AVs are collated in Table 1 [18]. Also included in this table are the reasons the WHO has given for not including a GV after calculating an HBV or health-based RV, or for establishing a GV that is higher (less protective) than its calculated HBV or health-based RV. This table also lists the health effects that are the basis for each HBV, as well as the year of publication of the POD study on which the calculated guideline is based.

A visual representation of the changes to the WHO SVs and GVs over time is shown in Fig 1. On this chart, it can be seen that while the GVs for many contaminants were often lowered from 1958–1993, beginning in 1998, some GVs began to be increased, while the GV for only one contaminant (Mn) has been decreased since 1993; the GV for Mn was withdrawn in 2011, then reinstated with a lower value in 2022 [6, 8, 16, 107].

Fig 2 highlights the publication years of the studies used as PODs for each of the inorganic contaminants with WHO GVs, HBVs, or AVs. When background documents did not identify a specific study as a POD, they were classified as "Unspecified date" and the publication year of the earliest document with the same value in the reference chain was used. That is, when background documents did not identify a specific POD but referred to an earlier reference document with the same value, the publication date of the earlier reference was used for this figure, and it is assumed that the value that appears in the reference is based on a study or studies published at some unspecified date prior to the publication date of the reference document. Notably, the majority of the WHO GVs, HBVs, or AVs are based on PODs that were published at least 15 years ago, and none are based on PODs published within the last seven years.

## National drinking water standards

We were able to find evidence of drinking water standards, guidelines, or regulations for 179 countries from our list of 195 countries. We were able to obtain copies of the standards or information about their contents for 178 of these countries; one country, Turkmenistan, was noted as having regulations but we were not able to obtain a copy or information about its contents. Of the remaining 16 countries on our list, we found evidence that standards do not exist in eight of them. Despite extensive searching, we were not able to uncover any information about standards in the remaining eight countries; for the purposes of the analyses below, we classified these countries as not having standards.

The evidence for the standards or lack of standards was primary (official government documents) for 158 countries and secondary (non-governmental reports, peer-reviewed journal articles, and academic theses or dissertations) for 29 countries. For the remaining eight countries, as noted above, we could find neither primary nor secondary evidence of existence or non-existence of standards.

Table 2 lists all 195 countries along with the publication dates of their most recent drinking water regulations, population, GDP/capita, World Bank income class, regulatory links to international organizations, the national entity responsible for the regulation, the number of inorganic substances regulated, and the type of evidence (primary, secondary, or none) that we found. A complete database of drinking water regulations for inorganic chemicals in each of these 195 countries is given in S1 File.

Fig 3 highlights the countries for which we could or could not find evidence of national drinking regulations.

As of 2020, the total population living in countries for which we could not find evidence of drinking water regulations was 151,997,043 people, or 2% of the worlds' total population [25].

**Table 1. The 2022 World Health Organization (WHO) drinking-water guideline values (GVs), health-based values (HBVs), reference values (RVs), and aesthetic values (AVs) for inorganic contaminants [18].** These values are in milligrams per liter (mg/L) and micrograms per liter (μg/L). Point of Departure (POD) studies are identified as provided in WHO background documents.

| Contaminant | GV | HBV or health-based RV | AV | Reason for not using HBV as GV | Health effects on which the HBV is based | Publication year(s) of POD study or studies |
|---|---|---|---|---|---|---|
| Aluminum (Al) | None | 0.9 mg/L (900 μg/L) | 0.1–0.2 mg/L (100–200 μg/L) | "Exceeds practicable levels based on opti-mization of the coagu-lation pro-cess in drinking-water plants using aluminium-based coagulants [18, 34].ᵃ" | Reproductive and neuro-develop-mental effects in mice, rats, and dogs [18, 35]. | <2007 References not provided [35]. |
| Ammonia (NH₃) | Noneᵇ | Noneᵇ | 1.5 mg/L (1,500 μg/L) | "Occurs in drinking-water at con-centrations well below those of health concern [18, 36]." | Reduced insulin sensitivity in unspecified types of subjects [36, 37]. | Effect and study not mentioned in cited source [36, 37]. |
| Antimony (Sb) | 0.02 mg/L (20 μg/L) | 0.02 mg/L (20 μg/L) | None | NA | Decreased body weight gain and food intake in rats [38–40]. | 1998, 1999 |
| Arsenic (As) | 0.01 mg/L (10 μg/L) (Pro-visional) | Not calculated | None | "Treatment performance" and "analytical achievability [18, 31]." | Cancer in humans [18, 31, 41]. | <2001 |
| Barium (Ba) | 1.3 mg/L (1,300 μg/L) | 1.3 mg/L (1,300 μg/L) | None | NA | Nephropathy in mice [18, 42, 43]. | 1994 |
| Beryllium (Be) | None | 0.012 mg/L (12 μg/L) | None | "Rarely found in drinking-water at concentra-tions of health concern [18]." | Gastro-intestinal lesions in dogs [18, 44–46]. | 1976 |
| Boron (B) | 2.4 mg/L (2,400 μg/L)ᶜ | 2.4 mg/L (2,400 μg/L) ᶜ | None | NA | Decreased fetal body weight in rats [18, 47, 48]. | 1996 |
| Cadmium (Cd) | 0.003 mg/L (3 μg/L) | 0.003 mg/L (3 μg/L) | None | NA | Biomarkers in human urine [18, 49, 50]. | <2011 References not provided [50]. |
| Chloride (Cl⁻) | None | None | 200–300 mg/L (200,000–300,000 μg/L) | "Not of health concern at levels found in drinking-water [18]." | Little is known about toxicity [51]. | References not provided [51]. |
| Chromium (Cr) | 0.05 mg/L (50 μg/L) | 0.05 mg/L (50 μg/L) | None | NA | Intestinal hyperplasia in mice [18, 52–54]. | 2008 |
| Copper (Cu) | 2 mg/L (2,000 μg/L) | 2 mg/L (2,000 μg/L) | 1–5 mg/L (1,000–5,000 μg/L) | NA | Acute gastro-intestinal effects in humans [18, 55–62] | 1998, 1999, 2001, 2003 |
| Cyanide (CN⁻) | None | Acute: 0.5 mg/L (500 μg/L) Chronic: 0.07 mg/L (70 μg/L) | None | "Occurs in drinking-water at concentrations well below those of health concern, except in emergency situations following a spill to a water source [18]." | Acute: Reproductive effects in rats with chronic exposures [18, 63, 64]. Chronic: Changes in behavior and serum biochemistry in pigs [65, 66]. | Acute: 1993. Chronic: 1988. |
| Fluoride (F⁻) | 1.5 mg/L (1,500 μg/L)ᵈ | 1.5 mg/L (1,500 μg/L)ᵈ | None | NA | Dental fluorosis in humans [18, 67]. | No calculations; no point of departure identified [67]. |
| Hydrogen Sulfide (H₂S) | None | None | 0.05–0.1 mg/L (50–100 μg/L) | NA | No data on oral toxicity [18, 68]. | NA |
| Iodine (I)ᵉ | None | Not calculated | None | Inadequate data and lifetime exposure unlikely [18]. | Thyroid effects possible in vulnerable populations [18, 69]. | No calculations; no point of departure identified [69]. |
| Iron (Fe) | None | 2 mg/L 2,000 μg/L | 0.3 mg/L (300 μg/L) | "Not of health concern at levels found in drinking water[18]." | Precaution against excess iron storage in humans [18, 70, 71]. | <1983 No calculations; no point of departure identified [71]. |

*(Continued)*

**Table 1.** (Continued)

| Contaminant | GV | HBV or health-based RV | AV | Reason for not using HBV as GV | Health effects on which the HBV is based | Publication year(s) of POD study or studies |
|---|---|---|---|---|---|---|
| **Lead (Pb)** | 0.01 mg/L (10 µg/L) (Pro-visional) | Cannot be calculated due to lack of threshold | None | Treatment and analytical achievability [18, 72]. | Neuro-develop-mental effects in humans [18, 72–74]. | Guideline based on treatment limitations; no references provided for treatment limitations [72] |
| **Manganese (Mn)** | 0.08 mg/L (80 µg/L) (Pro-visional) | 0.08 mg/L (80 µg/L) | 0.02 mg/L (20 µg/L) | NA | Neuro-developmental effects in neonatal rats [75–79]. | 1982 2010, 2011, 2013, 2017. |
| **Mercury (Hg)** | 0.006 mg/L (6 µg/L) Inorganic[f] | 0.006 mg/L (6 µg/L) Inorganic[f] | None | NA | Kidney changes in rats [18, 80, 81]. | 1993. |
| **Molybdenum (Mo)** | None | 0.07 mg/L (70 µg/L) | None | "Occurs in drinking-water at concentrations well below those of health concern [18]." | No urinary effects in humans [18, 82, 83]. | 1979. |
| **Nickel (Ni)** | 0.07 mg/L (70 µg/L) | 0.08 mg/L (80 µg/L) | None | The new (2021) HBV for chronic exposures is only slightly higher than the old GV of 0.07 mg/L [84]. | Reproductive and develop-mental toxicity in rats [18, 84–86][g]. | 2000[g]. |
| **Nitrate (NO₃⁻)** | 50 mg/L (50,000 µg/L) (short-term) | 50 mg/L (50,000 µg/L) | None | NA | Metha-hemoglobin-aemia in infants (short term) [18, 87]. | No calculations; no point of departure identified [87] |
| **Nitrite (NO₂⁻)** | 3 mg/L (3,000 µg/L) (short-term) | 3 mg/L (3,000 µg/L) | None | NA | Metha-hemoglobin-aemia in infants (short term) [18, 87]. | Not identified [87]. |
| **Potassium (K)** | None | Not calculated | None | NA | Kidney damage in susceptible humans [18, 88]. | <2009 Not identified [18, 88]. |
| **Selenium (Se)** | 0.04 mg/L (40 µg/L) (Pro-visional)[h] | 0.04 mg/L (40 µg/L) [h] | None | NA | Nail and hair effects in humans [18, 89–91]. | 1994 |
| **Silver (Ag)** | None | 0.1 mg/L (100 µg/L) (Pro-visional)[i] | None | "Available data in-adequate to permit derivation of health-based guideline value and usually occurs in drinking water at concentrations well below those of health concern [18, 32]." | Skin color changes in humans [18, 32, 92]. | 2009. |
| **Sodium (Na)** | None | None | 200 mg/L (200,000 µg/L) | NA | NA[j] | < 2003 No calculations; no point of departure identified [93]. |
| **Sulfate (SO₄²⁻)** | None | None[k] | 250–1,000 mg/L (250,000–1,000,000 µg/L) | "Not of health concern at levels found in drinking-water [18.]". | NA | NA |
| **Tin (Sn) (inorganic)** | None | Not calculated[l] | None | "Occurs in drinking-water at concentrations well below those of health Concern [18]." | Acute gastro-intestinal effects in humans [18, 94, 95]. | < 1989 References not available [95]. |
| **Uranium (U)** | 0.03 mg/L (30 µg/L) (Pro-visional)[m] | 0.03 mg/L (30 µg/L)[m] | None | NA | No kidney effects in humans [18, 96, 97]. | 2006. |

(*Continued*)

**Table 1.** (Continued)

| Contaminant | GV | HBV or health-based RV | AV | Reason for not using HBV as GV | Health effects on which the HBV is based | Publication year(s) of POD study or studies |
|---|---|---|---|---|---|---|
| **Zinc (Zn)** | None | Not calculated[n] | 3–5 mg/L (3,000–5,000 µg/L) | "Not of health concern at levels found in drinking-water [18]." | Not specified [18, 98, 99]. | < 1982 References not provided [99]. |

Abbreviations: AV: Aesthetic Value; GV: Guideline Value; HBV: Health-based Value; NA: Not available; POD: Point of Departure; RV: Reference Value

[a] The reason given for not establishing a GV only applies to Al that is found in water due to the use of Al compounds for treatment. Some water sources exceed the HBV even if no Al compounds are used for treatment; a GV would be useful for such cases [6].

[b] According to the WHO background document for NH₃, "With ammonium chloride, the acidotic effects of the chloride ion seem to be of greater importance than those of the ammonium ion. At a dose of more than 100 mg/kg of body weight per day (33.7 mg of ammonium ion per kg of body weight per day), ammonium chloride influences metabolism by shifting the acid–base equilibrium, disturbing the glucose tolerance, and reducing the tissue sensitivity to insulin [36]." The WHO did not develop a TDI from this data [36]. In the chemical factsheets section of the drinking water guidelines, the WHO states, "Toxicological effects are observed only at exposures above about 200 mg/kg body weight [18]."

[c] It should be noted that this value is based on a derivation that rounded the TDI at an intermediate step, resulting in an inflated HBV and GV. Following standard WHO procedures for calculating guidelines, which specify that there should be no rounding at intermediate steps of the derivations, a value of 2 mg/L would be expected for both the HBV and the GV rather than 2.4 mg/L [6].

[d] The 1.5 mg/L WHO health based drinking water guideline for F⁻ was first established in 1984 without a formal derivation [67]. The WHO noted that dental fluorosis was observed in a Chinese study in humans for 1 mg/L for 46% of the population examined [67, 100], putting into question whether the current 1.5 mg/L WHO GV for F⁻ is sufficiently low to protect public health.

[e] "Iodine [I] occurs naturally in water in the form of iodide [I⁻] [18]."

[f] It should be noted that this value is based on the toxicity of specifically inorganic Hg; thus, the guideline assumes that there will be no exposure to the more toxic methylmercury through drinking water, so no guidance is needed for methylmercury. However, if this value is used to evaluate total Hg or exposures to water containing methylmercury, it is too high to provide sufficient protection [6].

[g] The POD studies were unpublished and industry-funded [85, 86]. A 2015 Interlibrary Loan request for these papers was denied by the funder, the Nickel Producers Environmental Research Association [6, 101].

[h] It should be noted that this value is based on adult exposures and may not be sufficient to protect children from chronic exposures to water with high concentrations of Se [6].

[i] The 0.1 mg/L (100 µg/L) provisional guideline for Ag is termed a "reference value" or a "bounding value" rather than a formal health-based value [8, 32]. The WHO notes, "The toxicological database on silver is not adequate to support derivation of a formal guideline value. Nevertheless, it is recognized that a "bounding value" may be useful. Striking diffuse, blue–grey skin discolouration was reported in a woman who ingested 1 L (about 34 mg) of colloidal silver per day for approximately 16 months (0.6 mg/kg bw/day, assuming a body weight of 60 kg; Kim et al., 2009). This may be the lowest chronic LOAEL [Lowest Observed Adverse Effect Level] in humans, although it may be viewed as an aesthetic end-point rather than a health-related end-point. An uncertainty factor of 100 (10 for intraspecies variability and 10 for limited data, including both the short duration of the study, uncertainty associated with a dose level derived from human recall, and use of a LOAEL) applied to the LOAEL of 0.6 mg/kg bw/day, with an allocation factor of 80% and intake of 2 L of drinking-water daily, and 60 kg body weight, results in a drinking-water concentration of approximately 0.1 mg/L. This concentration can be considered the provisional health-based reference value for silver in drinking-water (i.e. maximum allowable concentration) [32, 92]."

[j] An independent WHO publication proposes a maximum intake of 2 g of Na/day [102]. Using the standard WHO assumptions (an adult drinking 2 L of water per day with a 20% exposure allocation factor to water), an HBV of 200 mg of Na/L in drinking water could be estimated from this recommendation as follows:

$$\left( \frac{2 \text{ g of Na}}{\text{day}} \times \frac{1{,}000 \text{ mg of Na}}{1 \text{ g of Na}} \times \frac{1 \text{ day}}{2 \text{ L of drinking water}} \right) \times 20\% = \frac{200 \text{ mg of Na}}{\text{L of drinking water}} \quad (i)$$

The publication years of the point of departure studies are 1984, 1989, 1992, 1993, 1997, 2001, 2009 [102]. The recommendation for a maximum intake of 2 g of Na/day was not a formal TDI; it was a compromise between reducing adverse health effects from ingesting too much Na and maintaining dietary palatability, and many diets exceed this intake from food alone [102]. With respect to drinking water, the WHO notes "No health-based guideline value has been derived, as the contribution from drinking-water to daily intake is small [18]." However, when the contribution of water to daily intake is small with other contaminants such as Hg, the approach was to lower the allocation factor, which results in a lower HBV, rather than to decline to set an HBV altogether [80]. Scheelbeek et al. reported that drinking water Na concentrations were highly associated with human blood pressure with water containing less than 200 mg of Na/L of drinking water [103].

[k] Gastrointestinal effects in pigs and humans are observed with SO₄²⁻ in excess of 500–1,200 mg/L of drinking water (500,000–1,200,000 µg/L) [104–106]. The publication years of the health effects studies are 1995 and 1997. The WHO states, "No health-based guideline is proposed for sulfate. However, because of the gastrointestinal effects resulting from ingestion of drinking-water containing high sulfate levels, it is recommended that health authorities be notified of sources of drinking water that contain sulfate concentrations in excess of 500 mg/l [18]."

[l] Using the standard WHO assumptions (a 60 kg adult drinking 2 L per day with 20% allocation factor) and a TDI of 2 mg of Sn/kg of body weight/day for acute exposures [18, 94, 95], an HBV of 12 mg of Sn/L of drinking water could be estimated for acute exposures as follows:

$$\left( 60 \text{ kg of body weight} \times \frac{2 \text{ mg of Sn}}{\text{kg of body weight} \times \text{day}} \times \frac{1 \text{ day}}{2 \text{ L of drinking water}} \right) \times 20\% = \frac{12 \text{ mg of Sn}}{\text{L of drinking water}} \quad (i)$$

[m] It should be noted that this calculation is based on the WHO's assertion that the population in the study in the POD was a no-effect group. However, this population showed statistically significant increases in diastolic blood pressure, systolic blood pressure, and glucose excretion in urine, which puts into question the WHO's assumption that no adverse effects were observed in this population [6, 97].

[n] Using the standard WHO assumptions (a 60 kg adult drinking 2 L per day with a 20% allocation factor) and a PMTDI (Provisional Maximum Tolerable Daily Intake) of 1 mg of Zn/kg of body weight/day [18, 98, 99], an HBV of 6 mg of Zn/L of drinking water (6,000 µg/L) could be estimated for acute exposures as follows:

$$\left( 60 \text{ kg of body weight} \times \frac{1 \text{ mg of Zn}}{\text{kg of body weight} \times \text{day}} \times \frac{1 \text{ day}}{2 \text{ L of drinking water}} \right) \times 20\% = \frac{6 \text{ mg of Zn}}{\text{L of drinking water}} \quad (i)$$

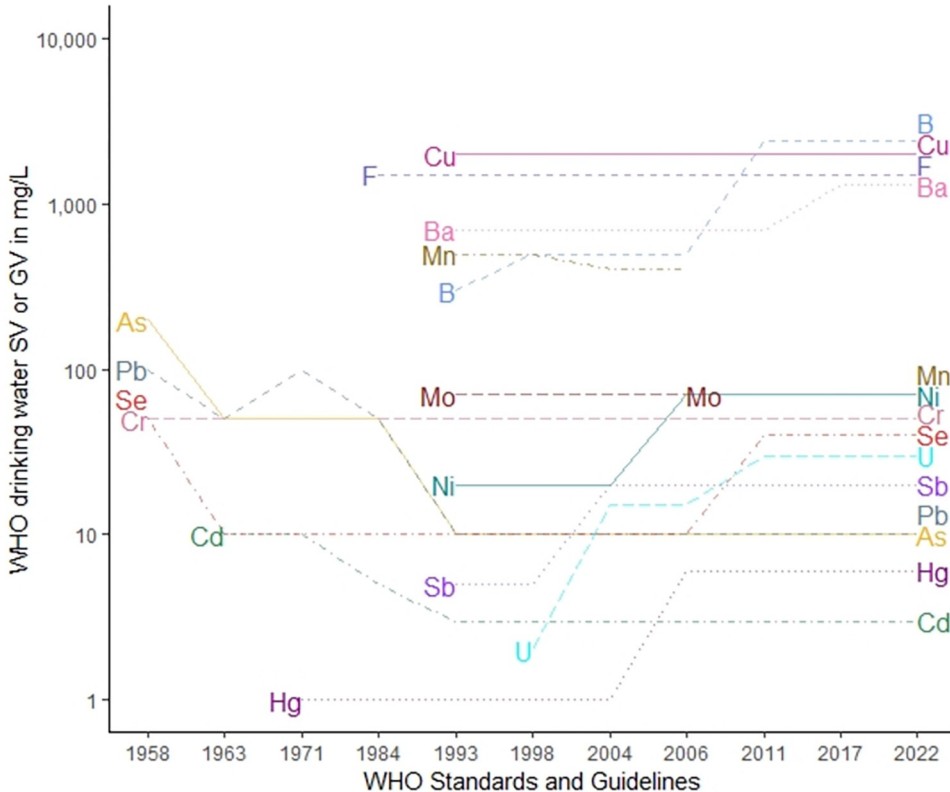

**Fig 1. A chronology of World Health Organization (WHO) standard values (SVs) and guidelines values (GVs) for selected inorganic contaminants [7–18].**

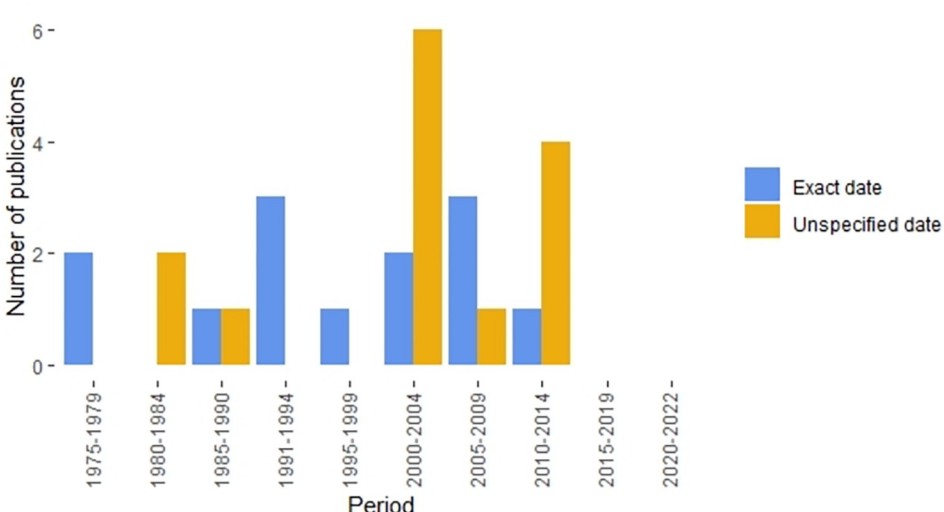

**Fig 2. Frequency of publication years for the point of departure (POD) studies or other references identified as the bases for World Health Organization guideline values (GVs), health-based values (HBVs), reference values (RVs), and aesthetic values (AVs) [7–18].**

**Table 2. Status of drinking water regulations for the 195 countries in this study.** For each country, information, when available, is listed for the number of inorganic chemicals regulated, the of inorganic chemical regulations exceeding World Health Organization (WHO) guideline values (GVs) [18], the year of publication of the regulations, the population in 2020 [25], the gross domestic product (GDP)/capita in United States dollars (USD) [26], the World Bank (WB) income class [108], regulatory links to other organizations or countries, the type of governmental body that released the regulation, and the type of evidence that we found for the regulation.

| Country and reference(s) | Number of inorganic chemicals regulated | Number of standards exceeding WHO GVs | Year of publication of these national regulations | Population in 2020 [25] | GDP/Capita in USD (2020) [26] | Income class[a] [108] | Regulatory affiliation or link | Regulation type | Evidence type |
|---|---|---|---|---|---|---|---|---|---|
| Afghanistan [109] | 24 | 2 | 2013 | 38,928,341 | $517 | Low | | Ministry | Primary |
| Albania [110] | 26 | 3 | 2016 | 2,837,743 | $5,246 | Upper-middle | | Legislation | Primary |
| Algeria [111] | 38 | 1 | 2011 | 43,851,043 | $3,307 | Lower-middle | | Legislation | Primary |
| Andorra [112] | 21 | 1 | 2007 | 77,265 | NA | High | | Legislation | Primary |
| Angola [113] | 15 | 3 | 2011 | 32,866,268 | $1,776 | Lower-middle | | Legislation | Primary |
| Antigua and Barbuda [114, 115] | 21 | 1 | 2003 | 97,928 | $13,993 | High | CARICOM | Legislation | Primary |
| Argentina [116] | 22 | 4 | 2019 | 45,376,763 | $8,579 | Upper-middle | | Legislation | Primary |
| Armenia [117, 118] | 9 | 2 | 2005 | 2,963,234 | $4,266 | Upper-middle | | Legislation | Secondary |
| Australia [119] | 32 | 3 | 2017 | 25,687,041 | $51,693 | High | | Ministry | Primary |
| Austria [120] | 21 | 1 | 2001 | 8,917,205 | $48,587 | High | EU | Legislation | Primary |
| Azerbaijan [121, 122] | 27 | 4 | 1985 | 10,093,121 | $4,221 | Upper-middle | CIS[b] | Bureau | Primary |
| Bahamas [115, 123] | 22 | 2 | 2010 | 393,248 | $25,194 | High | CARICOM | Bureau | Primary |
| Bahrain [124, 125] | 25 | 3 | 2012 | 1,701,583 | $20,410 | High | GCC | Bureau | Primary |
| Bangladesh [126] | 34 | 5 | 2019 | 164,689,383 | $1,962 | Lower-middle | | Ministry | Primary |
| Barbados [115, 127] | 16 | 0 | 2017 | 287,371 | $15,374 | High | CARICOM | Ministry | Primary |
| Belarus [128] | 44 | 6 | 2015 | 9,379,952 | $6,424 | Upper-middle | | Legislation | Primary |
| Belgium [129] | 24 | 1 | 2003 | 11,555,997 | $45,159 | High | EU | Legislation | Primary |
| Belize [115, 130] | 0 | NA | | 397,621 | $4,115 | Lower-middle | CARICOM | | Primary |
| Benin [131] | 21 | 6 | 2001 | 12,123,198 | $1,291 | Lower-middle | | Legislation | Primary |
| Bhutan [132] | 18 | 1 | 2018 | 771,612 | $3,001 | Lower-middle | | Ministry | Primary |
| Bolivia [133] | 25 | 1 | 2018 | 11,673,029 | $3,133 | Lower-middle | | Ministry | Primary |
| Bosnia and Herzegovina [134] | 21 | 1 | 2010 | 3,280,815 | $6,080 | Upper-middle | | Legislation | Primary |
| Botswana [135–137][c] | 25 | 0 | 2009 | 2,351,625 | $6,405 | Upper-middle | | Bureau | Secondary |
| Brazil [138] | 23 | 2 | 2021 | 212,559,409 | $6,797 | Upper-middle | | Legislation | Primary |
| Brunei [11, 139] | 25 | 1 | 1993 | 437,483 | $27,443 | High | WHO: 1993 | Ministry | Primary |
| Bulgaria [140] | 25 | 1 | 2001 | 6,934,015 | $10,079 | Upper-middle | EU | Legislation | Primary |
| Burkina Faso [141] | 25 | 1 | 2005 | 20,903,278 | $858 | Low | | Legislation | Primary |
| Burundi [142, 143] | 25 | 1 | 2000 | 11,890,781 | $239 | Low | EAS: 2000 | Bureau | Primary |

*(Continued)*

**Table 2.** (Continued)

| Country and reference(s) | Number of inorganic chemicals regulated | Number of standards exceeding WHO GVs | Year of publication of these national regulations | Population in 2020 [25] | GDP/Capita in USD (2020) [26] | Income class[a] [108] | Regulatory affiliation or link | Regulation type | Evidence type |
|---|---|---|---|---|---|---|---|---|---|
| Cambodia [144] | 22 | 2 | 2004 | 16,718,971 | $1,544 | Lower-middle | | Ministry | Primary |
| Cameroon [18, 145] | 24 | 0 | 2007 | 26,545,864 | $1,537 | Lower-middle | WHO | Ministry | Secondary |
| Canada [146] | 24 | 4 | 2020 | 38,005,238 | $43,258 | High | | Ministry | Primary |
| Cape Verde [147] | 27 | 2 | 2004 | 555,988 | $3,064 | Lower-middle | | Legislation | Primary |
| Central African Republic [18, 148] | 24 | 0 | 2017 | 4,829,764 | $493 | Low | WHO | Legislation | Primary |
| Chad [149] | 28 | 1 | 2010 | 16,425,859 | $659 | Low | | Legislation | Primary |
| Chile [150] | 18 | 3 | 2007 | 19,116,209 | $13,232 | High | | Legislation | Primary |
| China [151] | 26 | 2 | 2006 | 1,410,929,362 | $10,435 | Upper-middle | | Bureau | Primary |
| Colombia [152] | 24 | 1 | 2007 | 50,882,884 | $5,335 | Upper-middle | | Ministry | Primary |
| Comoros [153] | 13 | 4 | 1994 | 869,595 | $1,421 | Lower-middle | | Legislation | Primary |
| Congo | NA | NA | | 5,518,092 | $1,846 | Lower-middle | | | None |
| Congo, Democratic Republic | NA | NA | | 89,561,404 | $544 | Low | | | None |
| Costa Rica [154] | 24 | 1 | 2015 | 5,094,114 | $12,141 | Upper-middle | | Legislation | Primary |
| Croatia [155] | 11 | 1 | 2019 | 4,047,200 | $14,134 | High | EU | Legislation | Primary |
| Cuba [156] | 22 | 4 | 2017 | 11,326,616 | $9,478 | Upper-middle | | Bureau | Primary |
| Cyprus [157] | 14 | 1 | 2001 | 1,207,361 | $20,385 | High | EU | Legislation | Primary |
| Czech Republic [158] | 26 | 1 | 2014 | 10,698,896 | $22,932 | High | EU | Legislation | Primary |
| Denmark [159] | 24 | 0 | 2015 | 5,831,404 | $61,063 | High | EU | Legislation | Primary |
| Djibouti [160] | 23 | 3 | 2001 | 988,002 | $3,425 | Lower-middle | | Legislation | Primary |
| Dominica [18, 115, 161] | 24 | 0 | 2017 | 71,991 | $7,004 | Upper-middle | WHO; CARICOM | Ministry | Secondary |
| Dominican Republic [162] | 26 | 5 | 2001 | 10,847,904 | $7,268 | Upper-middle | | Ministry | Primary |
| Ecuador [163] | 14 | 2 | 2015 | 17,643,060 | $5,600 | Upper-middle | | Legislation | Primary |
| Egypt [164, 165] | 23 | 1 | 2007 | 102,334,403 | $3,569 | Lower-middle | | Ministry | Secondary |
| El Salvador [166] | 23 | 2 | 2009 | 6,486,201 | $3,799 | Lower-middle | | Ministry | Primary |
| Equatorial Guinea | NA | NA | | 1,402,985 | $7,143 | Upper-middle | | | None |
| Eritrea | NA | NA | | 3,620,517 | NA | Low | | | None |
| Estonia [167] | 21 | 1 | 2002 | 1,331,057 | $23,027 | High | EU | Ministry | Primary |
| Eswatini (Swaziland) [168–171] | 26 | 1 | 2015 | 1,160,164 | $3,424 | Lower-middle | | Bureau | Primary |

(*Continued*)

**Table 2.** (*Continued*)

| Country and reference(s) | Number of inorganic chemicals regulated | Number of standards exceeding WHO GVs | Year of publication of these national regulations | Population in 2020 [25] | GDP/Capita in USD (2020) [26] | Income class[a] [108] | Regulatory affiliation or link | Regulation type | Evidence type |
|---|---|---|---|---|---|---|---|---|---|
| Ethiopia [172] | 25 | 1 | 2013 | 114,963,583 | $936 | Low | | Bureau | Primary |
| Fiji [173, 174] | 24 | 1 | 2011 | 896,444 | $5,058 | Upper-middle | | Legislation | Secondary |
| Finland [175] | 22 | 1 | 2014 | 5,530,719 | $48,773 | High | EU | Ministry | Primary |
| France [176, 177] | 22 | 1 | 2017 | 67,391,582 | $39,030 | High | EU | Legislation | Primary |
| Gabon [178] | 15 | 3 | 2011 | 2,225,728 | $6,882 | Upper-middle | | Ministry | Primary |
| Gambia, The [179] | 21 | 3 | 2008 | 2,416,664 | $773 | Low | EU | Ministry | Primary |
| Georgia [180] | 24 | 1 | 2014 | 3,714,000 | $4,267 | Upper-middle | | Legislation | Primary |
| Germany [181] | 22 | 0 | 2017 | 83,240,525 | $46,208 | High | EU | Legislation | Primary |
| Ghana [182–184] | 24 | 0 | 2017 | 31,072,945 | $2,206 | Lower-middle | | Bureau | Secondary |
| Greece [185] | 21 | 1 | 2017 | 10,715,549 | $17,623 | High | EU | Legislation | Primary |
| Grenada [115, 186] | 21 | 1 | 2005 | 112,519 | $9,262 | Upper-middle | CARICOM | Legislation | Primary |
| Guatemala [187] | 19 | 2 | 2013 | 16,858,333 | $4,603 | Upper-middle | | Ministry | Primary |
| Guinea [188] | 24 | 0 | 1997 | 13,132,792 | $1,194 | Low | WHO | Legislation | Primary |
| Guinea-Bissau | NA | NA | | 1,967,998 | $728 | Low | | | None |
| Guyana [115] | 0 | NA | | 786,559 | $6,956 | Upper-middle | CARICOM | | Primary |
| Haiti [115, 189] | 24 | 0 | 2017 | 11,402,533 | $1,272 | Lower-middle | WHO; CARICOM | Ministry | Primary |
| Honduras [190] | 25 | 1 | 2007 | 9,904,608 | $2,389 | Lower-middle | | Legislation | Primary |
| Hungary [191] | 21 | 2 | 2002 | 9,749,763 | $15,981 | High | EU | Legislation | Primary |
| Iceland [192] | 21 | 1 | 2001 | 366,425 | $59,270 | High | | Legislation | Primary |
| India [193] | 25 | 1 | 2012 | 1,380,004,385 | $1,928 | Lower-middle | | Bureau | Primary |
| Indonesia [194] | 25 | 2 | 2010 | 273,523,621 | $3,870 | Lower-middle | | Ministry | Primary |
| Iran [195] | 27 | 1 | 2010 | 83,992,953 | $2,422 | Lower-middle | | Bureau | Primary |
| Iraq [196, 197] | 26 | 3 | 2009 | 40,222,503 | $4,146 | Upper-middle | | Bureau | Secondary |
| Ireland [198] | 21 | 1 | 2014 | 4,994,724 | $85,268 | High | EU | Legislation | Primary |
| Israel [199] | 25 | 4 | 2016 | 9,216,900 | $44,169 | High | | Legislation | Primary |
| Italy [200, 201] | 22 | 1 | 2016 | 59,554,023 | $31,714 | High | EU | Ministry | Primary |
| Ivory Coast [18, 202] | 24 | 0 | 2017 | 26,378,275 | $2,326 | Lower-middle | WHO | Legislation | Primary |
| Jamaica [115, 203] | 0 | NA | | 2,961,161 | $4,665 | Upper-middle | CARICOM | | Primary |
| Japan [204] | 17 | 0 | 2015 | 125,836,021 | $40,193 | High | | Ministry | Primary |
| Jordan [205] | 25 | 1 | 2015 | 10,203,140 | $4,283 | Upper-middle | | Bureau | Primary |
| Kazakhstan [206] | 22 | 4 | 2015 | 18,754,440 | $9,122 | Upper-middle | | Legislation | Primary |

(*Continued*)

**Table 2.** (*Continued*)

| Country and reference(s) | Number of inorganic chemicals regulated | Number of standards exceeding WHO GVs | Year of publication of these national regulations | Population in 2020 [25] | GDP/Capita in USD (2020) [26] | Income class[a] [108] | Regulatory affiliation or link | Regulation type | Evidence type |
|---|---|---|---|---|---|---|---|---|---|
| Kenya [207–209] | 26 | 1 | 2018 | 53,771,300 | $1,879 | Lower-middle | | Bureau | Primary |
| Kiribati [3, 18] | 24 | 0 | 2017 | 119,446 | $1,654 | Lower-middle | WHO | Ministry | Secondary |
| Korea, North | NA | NA | | 25,778,815 | NA | Low | | | None |
| Korea, South [210, 211] | 18 | 1 | 2015 | 51,780,579 | $31,631 | High | | Ministry | Primary |
| Kosovo [212] | 28 | 4 | 2012 | 1,775,378 | $4,347 | Upper-middle | EU | Legislation | Primary |
| Kuwait [125, 213] | 22 | 1 | 2011 | 4,270,563 | $24,812 | High | | Ministry | Secondary |
| Kyrgyzstan [214] | 23 | 4 | 2004 | 6,591,600 | $1,174 | Lower-middle | | Legislation | Primary |
| Laos [215] | 23 | 2 | 2009 | 7,275,556 | $2,630 | Lower-middle | | Ministry | Primary |
| Latvia [216] | 21 | 1 | 2017 | 1,901,548 | $17,726 | High | EU | Legislation | Primary |
| Lebanon [217] | 24 | 2 | 1999 | 6,825,442 | $4,650 | Upper-middle | | Bureau | Primary |
| Lesotho [218] | 0 | NA | | 2,142,252 | $875 | Lower-middle | | | Secondary |
| Liberia [18, 219] | 24 | 0 | 2017 | 5,057,677 | $633 | Low | WHO | Ministry | Primary |
| Libya [220–222] | 22 | 1 | 2015 | 6,871,287 | $3,699 | Upper-middle | | Bureau | Secondary |
| Liechtenstein [223] | 21 | 1 | 2018 | 38,137 | NA | High | | Legislation | Primary |
| Lithuania [224] | 21 | 1 | 2017 | 2,794,700 | $20,234 | High | EU | Legislation | Primary |
| Luxembourg [225] | 21 | 1 | 2017 | 632,275 | $116,015 | High | EU | Legislation | Primary |
| Macedonia, North [226] | 28 | 1 | 2018 | 2,072,531 | $5,917 | Upper-middle | | Legislation | Primary |
| Madagascar [227] | 25 | 3 | 2004 | 27,691,019 | $471 | Low | | Legislation | Primary |
| Malawi [228–231] | 13 | 2 | 2013 | 19,129,955 | $637 | Low | | Bureau | Secondary |
| Malaysia [232] | 27 | 1 | 2004 | 32,365,998 | $10,412 | Upper-middle | | Ministry | Primary |
| Maldives [233] | 17 | 1 | 2017 | 540,542 | $6,924 | Upper-middle | | Ministry | Primary |
| Mali [234] | 26 | 1 | 2007 | 20,250,834 | $862 | Low | | Ministry | Primary |
| Malta [235] | 21 | 1 | 2009 | 525,285 | $27,885 | High | EU | Legislation | Primary |
| Marshall Islands [236] | 18 | 7 | 1994 | 59,194 | NA | Upper-middle | | Legislation | Primary |
| Mauritania [237] | 24 | 0 | 2015 | 4,649,660 | $1,702 | Lower-middle | WHO | Ministry | Primary |
| Mauritius [238] | 14 | 0 | 1996 | 1,265,740 | $8,628 | Upper-middle | | Legislation | Primary |
| Mexico [239] | 19 | 1 | 2019 | 128,932,753 | $8,329 | Upper-middle | | Legislation | Primary |
| Micronesia, Federated States [240, 241] | 23 | 7 | 2018 | 115,021 | $3,565 | Lower-middle | US | Legislation | Primary |
| Moldova [242] | 22 | 0 | 2019 | 2,620,495 | $4,547 | Upper-middle | | Legislation | Primary |
| Monaco [243] | 22 | 1 | 2017 | 39,244 | NA | High | | Legislation | Primary |

(*Continued*)

**Table 2.** (Continued)

| Country and reference(s) | Number of inorganic chemicals regulated | Number of standards exceeding WHO GVs | Year of publication of these national regulations | Population in 2020 [25] | GDP/Capita in USD (2020) [26] | Income class[a] [108] | Regulatory affiliation or link | Regulation type | Evidence type |
|---|---|---|---|---|---|---|---|---|---|
| Mongolia [244–246] | 11 | 1 | 2018 | 3,278,292 | $4,061 | Lower-middle | | Bureau | Primary |
| Montenegro [247] | 31 | 2 | 2012 | 621,306 | $7,677 | Upper-middle | | Legislation | Primary |
| Morocco [248–250] | 22 | 1 | 2006 | 36,910,558 | $3,108 | Lower-middle | | Bureau | Primary |
| Mozambique [251] | 27 | 1 | 2004 | 31,255,435 | $449 | Low | | Legislation | Primary |
| Myanmar [252–254] | 27 | 2 | 2014 | 54,409,794 | $1,468 | Lower-middle | | Ministry | Secondary |
| Namibia [255, 256] | 43 | 10 | 1988 | 2,540,916 | $4,179 | Upper-middle | | Legislation | Primary |
| Nauru [3, 18, 257][d] | 24 | 0 | 2017 | 10,834 | $10,580 | High | WHO | Ministry | Primary |
| Nepal [258] | 17 | 2 | 2005 | 29,136,808 | $1,155 | Lower-middle | | Ministry | Primary |
| Netherlands [259] | 23 | 1 | 2011 | 17,441,139 | $52,397 | High | EU | Legislation | Primary |
| New Zealand [260] | 26 | 4 | 2018 | 5,084,300 | $41,441 | High | | Ministry | Primary |
| Nicaragua [261] | 17 | 1 | 2000 | 6,624,554 | $1,905 | Lower-middle | | Legislation | Primary |
| Niger [18, 262] | 24 | 0 | 2017 | 24,206,636 | $568 | Low | WHO | Legislation | Primary |
| Nigeria [263] | 21 | 1 | 2015 | 206,139,587 | $2,097 | Lower-middle | | Bureau | Primary |
| Norway [264] | 21 | 1 | 2016 | 5,379,475 | $67,390 | High | | Legislation | Primary |
| Oman [125, 265] | 27 | 1 | 2012 | 5,106,622 | $12,660 | High | | Bureau | Primary |
| Pakistan [266] | 19 | 4 | 2010 | 220,892,331 | $1,189 | Lower-middle | | Ministry | Primary |
| Palau [267] | 10 | 4 | 1996 | 18,092 | $14,244 | High | | Ministry | Primary |
| Panama [268][e] | 20 | 1 | 2007 | 4,314,768 | $12,510 | Upper-middle | | Legislation | Primary |
| Papua New Guinea [269] | 17 | 6 | 2006 | 8,947,027 | $2,757 | Lower-middle | | Legislation | Primary |
| Paraguay [270] | 6 | 1 | 2000 | 7,132,530 | $5,001 | Upper-middle | | Legislation | Primary |
| Peru [271] | 27 | 1 | 2017 | 32,971,846 | $6,127 | Upper-middle | | Legislation | Primary |
| Philippines [272] | 23 | 1 | 2016 | 109,581,085 | $3,299 | Lower-middle | | Ministry | Primary |
| Poland [273] | 25 | 1 | 2017 | 37,950,802 | $15,721 | High | EU | Legislation | Primary |
| Portugal [274] | 21 | 1 | 2017 | 10,305,564 | $22,176 | High | EU | Legislation | Primary |
| Qatar [124, 275] | 20 | 1 | 2014 | 2,881,060 | $50,124 | High | | Ministry | Secondary |
| Romania [276] | 23 | 1 | 2019 | 19,286,123 | $12,896 | Upper-middle | EU | Legislation | Primary |
| Russia [277] | 22 | 4 | 2001 | 144,104,080 | $10,295 | Upper-middle | | Ministry | Primary |
| Rwanda [278, 279] | 26 | 1 | 2014 | 12,952,209 | $798 | Low | | Bureau | Secondary |
| Saint Kitts and Nevis [115] | 0 | NA | | 53,192 | $18,438 | High | CARICOM | | Primary |
| Saint Lucia [115] | 0 | NA | | 183,629 | $8,805 | Upper-middle | CARICOM | | Primary |

(Continued)

**Table 2.** (Continued)

| Country and reference(s) | Number of inorganic chemicals regulated | Number of standards exceeding WHO GVs | Year of publication of these national regulations | Population in 2020 [25] | GDP/Capita in USD (2020) [26] | Income class[a] [108] | Regulatory affiliation or link | Regulation type | Evidence type |
|---|---|---|---|---|---|---|---|---|---|
| Saint Vincent and the Grenadines [115] | 0 | NA | | 110,947 | $7,278 | Upper-middle | CARICOM | | Primary |
| Samoa [280–282] | 8 | 0 | 2016 | 198,410 | $4,068 | Lower-middle | | Ministry | Secondary |
| San Marino [283] | 15 | 3 | 2012 | 33,938 | NA | High | | Legislation | Primary |
| São Tomé and Príncipe [284] | 0 | NA | | 219,161 | $2,158 | Lower-middle | | | Secondary |
| Saudi Arabia [125, 285] | 27 | 1 | 2015 | 34,813,867 | $20,110 | High | | Ministry | Primary |
| Senegal [14, 286–289] | 26 | 1 | 1996 | 16,743,930 | $1,472 | Lower-middle | | Ministry | Secondary |
| Serbia [290] | 26 | 0 | 2019 | 6,908,224 | $7,721 | Upper-middle | | Legislation | Secondary |
| Seychelles [291] | 7 | 0 | 2012 | 98,462 | $10,764 | High | | Legislation | Primary |
| Sierra Leone [292] | 7 | 1 | 2019 | 7,976,985 | $509 | Low | | | Primary |
| Singapore [293] | 18 | 1 | 2019 | 5,685,807 | $59,798 | High | | Legislation | Primary |
| Slovakia [294] | 26 | 1 | 2006 | 5,458,827 | $19,267 | High | EU | Legislation | Primary |
| Slovenia [295] | 21 | 1 | 2015 | 2,100,126 | $25,517 | High | EU | Legislation | Primary |
| Solomon Islands [18, 296] | 26 | 4 | 2017 | 686,878 | $2,251 | Lower-middle | | Legislation | Primary |
| Somalia | NA | NA | | 15,893,219 | $314 | Low | | | None |
| South Africa [168, 169] | 26 | 1 | 2015 | 59,308,690 | $5,656 | Upper-middle | | Legislation | Primary |
| Spain [297] | 21 | 1 | 2003 | 47,351,567 | $27,063 | High | EU | Legislation | Primary |
| Sri Lanka [298] | 18 | 4 | 2019 | 21,919,000 | $3,681 | Lower-middle | | Legislation | Primary |
| Sudan [299, 300] | 20 | 1 | 2009 | 43,849,269 | $486 | Low | | Bureau | Secondary |
| Sudan, South [301] | 13 | 3 | 2011 | 11,193,729 | NA | Low | | Ministry | Primary |
| Suriname [114, 302] | 0 | NA | 2019 | 586,634 | $4,917 | Upper-middle | CARICOM | | Secondary |
| Sweden [303] | 23 | 1 | 2017 | 10,353,442 | $52,274 | High | EU | Legislation | Primary |
| Switzerland [304] | 25 | 0 | 2020 | 8,636,896 | $87,097 | High | | Ministry | Primary |
| Syria [305–307] | 22 | 4 | 2007 | 17,500,657 | NA | Low | | Bureau | Secondary |
| Taiwan [308] | 24 | 2 | 2017 | 23,816,775 | NA | | | Ministry | Primary |
| Tajikistan [122, 309] | 16 | 3 | 1982 | 9,537,642 | $859 | Lower-middle | CIS | Bureau | Secondary |
| Tanzania [310–312] | 28 | 1 | 2018 | 59,734,213 | $1,045 | Lower-middle | EAS | Bureau | Secondary |
| Thailand [313, 314] | 14 | 4 | 2008 | 69,799,978 | $7,187 | Upper-middle | | Ministry | Primary |
| Timor-Leste [18, 315] | 24 | 0 | 2017 | 1,318,442 | $1,443 | Lower-middle | WHO | Ministry | Secondary |
| Togo [316] | 23 | 1 | 2015 | 8,278,737 | $915 | Low | | Ministry | Primary |
| Tonga [3, 18, 317][f] | 24 | 0 | 2017 | 105,697 | $4,625 | Upper-middle | WHO | Ministry | Secondary |
| Trinidad and Tobago [115] | NA | NA | | 1,399,491 | $15,426 | High | CARICOM | | None |

*(Continued)*

**Table 2.** (Continued)

| Country and reference(s) | Number of inorganic chemicals regulated | Number of standards exceeding WHO GVs | Year of publication of these national regulations | Population in 2020 [25] | GDP/Capita in USD (2020) [26] | Income class[a] [108] | Regulatory affiliation or link | Regulation type | Evidence type |
|---|---|---|---|---|---|---|---|---|---|
| Tunisia [318] | 26 | 2 | 2013 | 11,818,618 | $3,522 | Lower-middle | | Bureau | Primary |
| Turkey [319, 320] | 27 | 2 | 2019 | 84,339,067 | $8,536 | Upper-middle | | Legislation | Primary |
| Turkmenistan [321–323] | ? | ? | 2016 | 6,031,187 | NA | Upper-middle | | Bureau | Primary |
| Tuvalu [18, 324] | 24 | 0 | 2017 | 11,792 | $4,143 | Upper-middle | WHO | Ministry | Primary |
| Uganda [325] | 26 | 1 | 2014 | 45,741,000 | $822 | Low | EAS | Bureau | Primary |
| Ukraine [326] | 28 | 1 | 2010 | 44,134,693 | $3,523 | Lower-middle | | Ministry | Primary |
| United Arab Emirates [125, 327][g] | 25 | 1 | 2014 | 9,890,400 | $36,285 | High | | Ministry | Secondary |
| United Kingdom [328][h] | 21 | 1 | 2016 | 67,215,293 | $41,125 | High | | Legislation | Primary |
| United States [241] | 23 | 7 | 2018 | 329,484,123 | $63,414 | High | | Ministry | Primary |
| Uruguay [329] | 28 | 3 | 2010 | 3,473,727 | $15,438 | High | | Bureau | Primary |
| Uzbekistan [330] | 21 | 4 | 2006 | 34,232,050 | $1,751 | Lower-middle | | Legislation | Primary |
| Vanuatu [331] | 17 | 1 | 2019 | 307,150 | $2,870 | Lower-middle | | Legislation | Primary |
| Venezuela [332] | 23 | 2 | 1998 | 28,435,943 | NA | | | Legislation | Primary |
| Vietnam [333] | 25 | 1 | 2009 | 97,338,583 | $2,786 | Lower-middle | | Ministry | Primary |
| Yemen [334] | 28 | 3 | 1999 | 29,825,968 | NA | Low | | Ministry | Primary |
| Zambia [335] | 23 | 2 | 2010 | 18,383,956 | $985 | Lower-middle | | Bureau | Primary |
| Zimbabwe [336–340] | 26 | 2 | 2014 | 14,862,927 | $1,215 | Lower-middle | | Bureau | Secondary |

Abbreviations: CARICOM: Caribbean Community; CIS: Commonwealth of Independent States; EAS: East African States; EU: European Union; GCC: Gulf Cooperation Council; GDP: Gross Domestic Product; GV: Guideline Value; NA: Not available or Not Applicable; US: United States; USD: United States Dollars; WB: World Bank; WHO: World Health Organization

[a] The WB 2022 income classes were based on the following ranges: low = < $1,045; lower-middle = $1,086–4,255; upper-middle = $4,256–13,205; high > $13,205 [108].

[b] The documentation for the standards for Azerbaijan state that they are derived from the Interstate Standards (GS 2874–82) [122]; however the regulations have been modified somewhat since the contents of the official Azerbaijan standard differ slightly from those of the original GS 2874–83 document [121, 122].

[c] Standard withdrawn October 14, 2016 [136].

[d] A Nauru government document states that there are no standards [257], but the 2018 WHO survey noted that Nauru uses WHO guidelines as standards [3, 18].

[e] The national standards apply specifically to bottled water [268].

[f] A secondary source states that there are no national standards [317], but the 2018 WHO survey noted that Tonga uses WHO guidelines as standards [3, 18].

[g] The standards are specifically for the Emirate of Abu Dhabi, one of the seven emirates [327].

[h] The standards apply specifically to private water supplies [328].

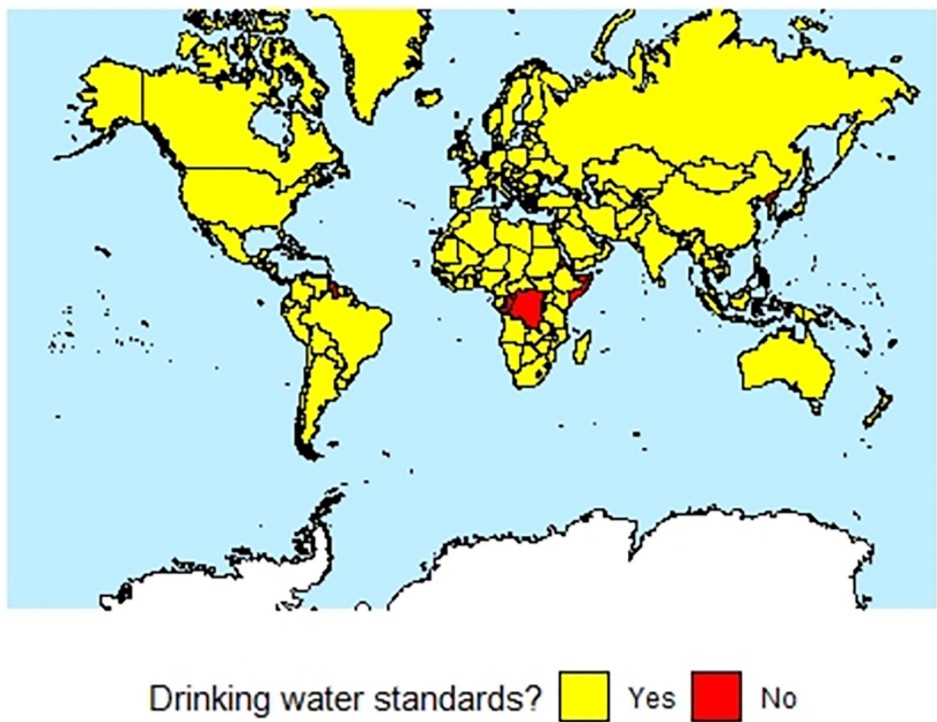

Drinking water standards? ☐ Yes ☐ No

**Fig 3. Map of 195 countries showing their drinking water regulation status (map base: [30]).**

Conversely, 7,591,923,366 people, or 98% of the world's population live in countries with established drinking water regulations (Table 2) [25].

On a national level, the median GDP/capita in countries with published national drinking water standards that we found and with World Bank GDP/capita data is $5,029, while the median GDP/capita in countries for which the World Bank has GDP/capita data but for which we could find no national drinking water standards GDP/capita data is $4,390. In contrast, the GDP/capita difference is more substantial when we focus on the populations living in countries with or without drinking water regulations. The P-W GDPs/capita of populations living in countries with drinking water regulations was $11,379, while the P-W GDPs/capita populations living in countries with no drinking water regulations was $1,008.

The average number of inorganic contaminants with regulations of the 178 countries whose regulations we were able to obtain was 22. Fig 4 highlights the 19 countries with regulations for 15 or fewer inorganic contaminants in drinking water, while Fig 5 highlights the 23 countries with regulations for 27 or more inorganic contaminants.

Regulations that specify values that are higher than the WHO GVs are less protective than the WHO GVs; we classified them as "exceedances". Of the 178 countries whose regulations we were able to obtain, 149 (84%) had regulations with three or fewer exceedances of WHO GVs. The remaining 28 countries (16%) had regulations containing more than three exceedances of WHO GVs. These countries are highlighted in Fig 6.

Of the countries shown in Fig 6, Namibia uses regulations proposed for South Africa in 1988 [255, 256]; South Africa's current regulations were published in 2015 [168, 169]. The Marshall Islands have drinking water regulations that were originally based on those of the United States, while Micronesia has regulations directly linked to those of the United States

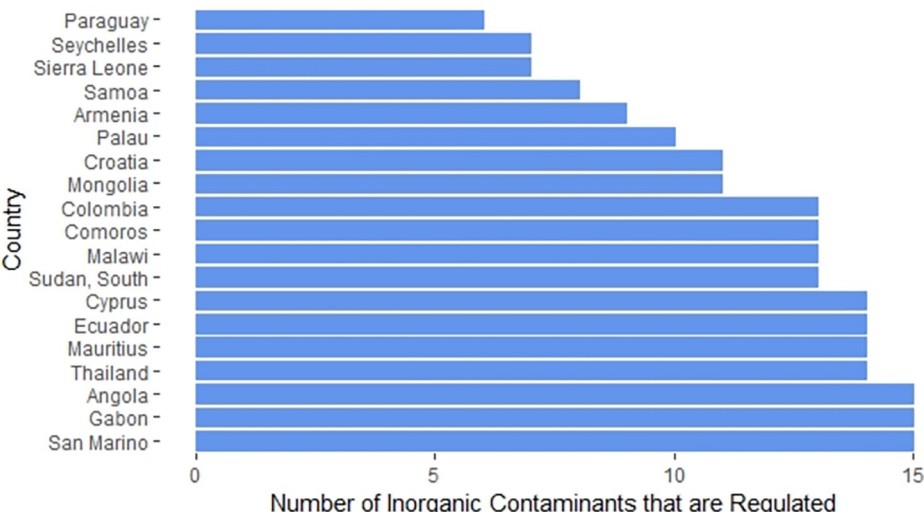

**Fig 4. Countries with regulations for 15 or fewer inorganic contaminants in drinking water.**

[236, 240]. The regulations for the remaining countries are independent; that is, they are not directly linked to the regulations of other countries or organizations.

## Recency of regulations

The oldest national regulations still in force go back to 1982, while the most recent regulations that we could find were updated in 2022. The modes of the publication dates were 2017 and 2022, while the mean was 2012 (Table 2).

## Regulations that are independent and regulations that are linked to guidelines or standards of other international organizations or entities

The European Union (EU) has established drinking water standards that individual member states have ratified through their national legislation. Twenty-eight countries use EU

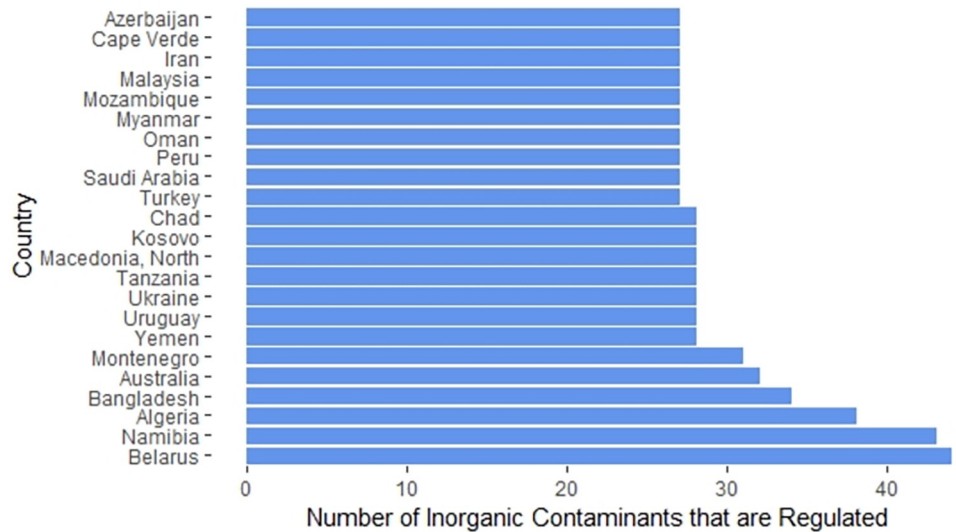

**Fig 5. Countries with regulations for 27 or more inorganic contaminants in drinking water.**

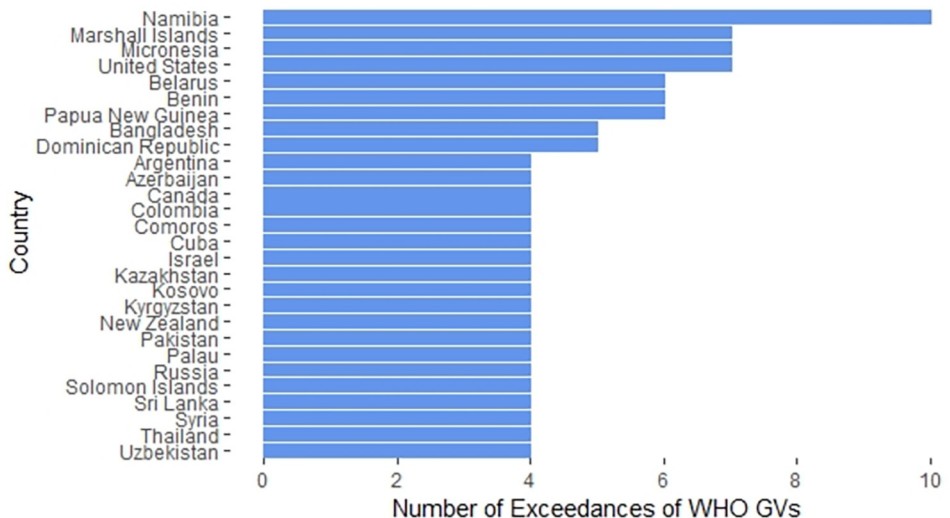

**Fig 6. Countries with more than three exceedances of WHO GVs for inorganic contaminants in their national drinking water regulations.** Regulations classified as exceedances provide less protection than WHO GVs.

regulations, or 6% of the world's total population. Fourteen countries (2% of the world's population) directly link their national standards to the current WHO's GVs. Other international organizations that have published drinking water standards used as national standards by one or more countries include the East African Standards (EAS) bureau (two countries, 1% of the world's population), the Commonwealth of Independent States (CIS; two countries, 0.2% of the world's population), and the Gulf Cooperation Council (GCC; one country, 0.02% of the world's population). In addition, the Caribbean Community (CARICOM) has published drinking water regulations for their 11 member nations; however, these regulations apply to bottled water only [115]. We were not able to find regulations for seven of the 11 CARICOM member nations, so the CARICOM bottled water regulations may be the only drinking water regulations applicable in these seven nations (0.08% of the world's population). Two countries (the United States and Micronesia) use standards published by the United States Environmental Protection Agency (U.S. EPA; two countries, 4% of the world's population). Drinking water regulations in the remaining 137 countries (87% of the world's population) are not linked to any international organizations or the standards of other countries.

## Government entities and transparency

National drinking water standards are created by a variety of government entities (Table 2). Some national standards are pronounced by presidential/royal decree or legislation (86 countries, 20% of world population) while in other countries, the drinking water standards are published by governmental agencies (57 countries, 30% of world population). Some countries rely on national standards agencies to establish and maintain drinking water standards (34 countries, 48% of world population). In some of these countries, the national standards agencies are part of the government, while in other countries the standards agencies are independent entities, not part of the government. In most countries that rely on standards agencies to develop and publish drinking water standards, the national standards agencies sell their standards to generate income (31 countries, 29% of world population). In these countries, the drinking water standards are copyrighted documents and can only be accessed through payments to the national standards agencies. An additional challenge for obtaining these standards documents

is that many of them can only be purchased in person at the national standards agencies' head-quarters. This is a barrier to transparency, since the citizens of these countries are unable to freely access the standards that apply to their drinking water.

On a national level, the GDPs/capita of countries whose national drinking water standards can be accessed without cost is higher than that of countries where there is a fee to access national drinking water standards. The median GDP/capita in countries where national drinking water standards can be accessed for free and the World Bank has GDP/capita data is $6,797, while the median GDP/capita in countries where national drinking water standards must be purchased for a fee from standards agencies and the World Bank has GDP/capita data is $2,206 [26]. However, the differences are more substantial when considering these countries on a P-W GDP/capita basis. The P-W GDP/capita of populations of countries in which national drinking water standards can be accessed without payments to standards agencies is $15,032, while the P-W GDP/capita of populations of countries in which national drinking water standards must be purchased from standards agencies is $1,956.

## Typographical errors

Typographical errors in drinking water guidelines and standards are generally rare but certainly can be found. For example, the summary table in the original printed version of the 4th edition of *Guidelines for Drinking-water Quality* by the WHO contained a typographical error. More specifically, the 2011 printed edition listed the guideline for U as both 30 μg/L and 0.30 mg/L [16]. The 30 μg/L value was the most likely intended value since it was consistent with the 30 μg/L listed in the chemical fact sheet (Chapter 12) for U. The second value listed for U, 0.30 mg/L, was supposed to be equivalent to the first, but was off by a factor of 10 (0.30 mg/L = 300 μg/L); the intended second value was most likely 0.030 mg/L [16, 341].

When examining the national drinking water regulations, we occasionally noted values that seemed to be typographical errors. We classified values as "potential errors" when they used the same numerals as the more common values of standards used by other countries or the WHO for a given contaminant but were at least one order of magnitude different from the mode. Often, these potential errors stood out as being the only values in a nation's standards that were not consistent with the WHO GVs. We found such potential typographical errors in the documentation for the national guidelines of 22 countries, representing 12% of the 179 countries for which we found evidence of standards (S1 File). We found potential typographical errors in different types of documents, including the documents of record (national legislative journals) and other official government documents, as well as in peer-reviewed articles (as described above, peer-reviewed articles were used as references for this database only when official government publications were unobtainable).

The median GDP/capita in countries where national drinking water standards in official government documents contained no potential errors and the World Bank has GDP/capita data for 2020 is $6,797, while the median GDP/capita in countries where national drinking water standards contained potential errors in official government documents and the World Bank has GDP/capita data for 2020 is $4,603. In contrast, when the data were weighted by population, the P-W GDPs/capita between countries whose national standards contained potential errors in government documents and those that did not was more substantial. The P-W GDP/capita of populations for countries with no potential errors in the national standards published in official government documents was $12,278, while the P-W GDP/capita for countries with at least one potential error in the national standards published in an official government document was $3,418.

Most of these potential errors seemed to be random, occurring only zero to six times for a given contaminant. Since these potential typographical errors were one or more orders

of magnitude different from the other values, they greatly increased the ranges (minima and maxima) for the affected contaminants. To gain more accurate insight into the ranges for the values intended by the legal standards, we tabulated separate minima and maxima for the entire data set as well as the subset of the data after potential typographical errors were set aside. However, the values of potential typographical errors in non-primary evidence (e.g. peer-reviewed papers) were deleted; these substances were marked "regulated" but the potentially erroneous values were not included in ranges or statistical calculations.

Two contaminants, $Cu$ and $NO_2^-$, had more than 10 potential typographical errors each when examined using the criteria of highlighting values that had similar numerals but different orders of magnitude from the mode. This pattern of potential errors concentrated in the values for $Cu$ and $NO_2^-$ did not seem to be random but more likely an indication of a large degree of variation in the national standards for these two contaminants. With such variation, it was impossible to identify which values were potential errors and which were accurate indications of the intended values, so we did not classify any of the values for $Cu$ and $NO_2^-$ as typographical errors.

## Categories of regulations

The drinking water standards for many countries make a formal distinction between contaminants that have known adverse health effects and contaminants for which the effects on drinking water quality are aesthetic in nature (e.g. unpleasant taste or odor or the staining of plumbing fixtures or laundry). In countries that make this distinction, the standards for contaminants that adversely affect health are often labelled "mandatory", while the standards for contaminants linked to aesthetic effects are typically labelled "indicators". Worldwide, 110 countries (61% of countries with regulations, 55% of the world's population) maintain these two categories of standards. In contrast, 69 countries (39% of countries with regulations, 43% of world population) do not make this distinction and term all their drinking water standards "mandatory".

There are a variety of legal definitions for the "indicator" contaminants. In some countries water suppliers are required to provide water that does not exceed "indicator" contaminant values, while in other countries, "indicator" contaminant values are simply goals to be strived for. Thirty-five countries (20% of countries with regulations, 12% of world population) provide both "mandatory" and "indicator" values for certain contaminants; in these countries, it may be the case that only the "mandatory" values are legal requirements for drinking water providers.

Four countries (2% of the world's population) specify exceptional values for specific contaminants to be used in "emergency" situations. These values are generally described as being for short-term exposures only. Two countries (18% of world population) also provide exceptional values for specific contaminants when no alternative water sources are available. Such situations are not time-limited, so exposures in these cases may be chronic.

Table 3 provides a list of all inorganic contaminants for which we found national drinking water standards in at least one country. This table also summarizes the number of countries that provide "mandatory" standards for each contaminant, the number of countries that provide "indicator" values, the number of countries that provide both "mandatory" and "indicator" values, the number of countries that provide values for situations when no alternative source is available, and the number of countries that provide values for "emergency" situations.

**Table 3. List of inorganic drinking water contaminants that are regulated by individual countries, along with the types of regulations and numbers of countries that have each type of regulation.**

| Contaminant | Mandatory | Indicator | Both Mandatory & Indicator | No Alternative | Emergency |
|---|---|---|---|---|---|
| Aluminum (Al) | 56 | 92 | 2 | 1 | 1 |
| Ammonia (NH$_3$) | 27 | 37 | 0 | 0 | 2 |
| Ammonium (NH$_4$$^+$) | 24 | 41 | 1 | 0 | 0 |
| Antimony (Sb) | 115 | 0 | 0 | 0 | 2 |
| Arsenic (As) | 175 | 0 | 0 | 1 | 2 |
| Barium (Ba) | 102 | 2 | 0 | 0 | 2 |
| Beryllium (Be) | 20 | 0 | 0 | 0 | 1 |
| Bicarbonate (HCO$_3$$^-$) | 1 | 0 | 0 | 0 | 0 |
| Bismuth (Bi) | 2 | 0 | 0 | 0 | 1 |
| Borate (BO$_3$$^{3-}$) | 1 | 0 | 0 | 0 | 0 |
| Boron (B) | 125 | 2 | 0 | 1 | 2 |
| Cadmium (Cd) | 167 | 0 | 0 | 0 | 1 |
| Calcium (Ca) | 26 | 26 | 0 | 1 | 1 |
| Cerium (Ce) | 1 | 0 | 0 | 0 | 1 |
| Chloride (Cl$^-$) | 61 | 104 | 0 | 1 | 2 |
| Chromium (Cr) | 169 | 0 | 0 | 0 | 2 |
| Cobalt (Co) | 12 | 1 | 0 | 0 | 1 |
| Copper (Cu) | 147 | 53 | 30 | 1 | 3 |
| Cyanide (CN$^-$) | 140 | 0 | 0 | 0 | 2 |
| Europium (Eu) | 0 | 1 | 0 | 0 | 0 |
| Fluoride (F$^-$) | 158 | 7 | 2 | 1 | 2 |
| Fluorine (F) | 1 | 0 | 0 | 0 | 0 |
| Gold (Au) | 1 | 0 | 0 | 0 | 1 |
| Hydrogen Sulfide (H$_2$S) | 12 | 32 | 0 | 0 | 1 |
| Hydrosulfide (HS$^-$) | 2 | 0 | 0 | 0 | 0 |
| Indium (In) | 1 | 0 | 0 | 0 | 0 |
| Iodide (I$^-$) | 2 | 0 | 0 | 0 | 1 |
| Iron (Fe) | 69 | 100 | 4 | 1 | 3 |
| Lanthanum (La) | 1 | 0 | 0 | 0 | 0 |
| Lead (Pb) | 172 | 0 | 0 | 0 | 2 |
| Lithium (Li) | 3 | 0 | 0 | 0 | 1 |
| Magnesium (Mg) | 35 | 27 | 0 | 1 | 2 |
| Manganese (Mn) | 101 | 90 | 28 | 1 | 3 |
| Mercury (Hg) | 166 | 0 | 0 | 0 | 2 |
| Molybdenum (Mo) | 43 | 0 | 0 | 0 | 2 |
| Nickel (Ni) | 140 | 1 | 0 | 0 | 2 |
| Niobium (Nb) | 1 | 0 | 0 | 0 | 0 |
| Nitrate (NO$_3$$^-$) | 168 | 8 | 1 | 0 | 2 |
| Nitrite (NO$_2$$^-$) | 142 | 7 | 2 | 0 | 1 |
| Nitrogen (N) | 7 | 3 | 0 | 0 | 0 |
| Dissolved oxygen (O$_2$) | 10 | 5 | 0 | 0 | 1 |
| Phosphate (PO$_4$$^{3-}$)[a] | 21 | 3 | 0 | 0 | 0 |
| Phosphorus (P) | 13 | 2 | 0 | 0 | 0 |
| Potassium (K) | 18 | 11 | 0 | 0 | 2 |
| Rhodium (Rh) | 1 | 0 | 0 | 0 | 0 |
| Rubidium (Rb) | 1 | 0 | 0 | 0 | 0 |

(*Continued*)

**Table 3.** (Continued)

| Contaminant | Mandatory | Indicator | Both Mandatory & Indicator | No Alternative | Emergency |
|---|---|---|---|---|---|
| Samarium (Sm) | 1 | 0 | 0 | 0 | 0 |
| Selenium (Se) | 160 | 0 | 0 | 0 | 2 |
| Silicone dioxide (SiO$_2$) | 1 | 0 | 0 | 0 | 0 |
| Silver (Ag) | 37 | 5 | 1 | 0 | 1 |
| Sodium (Na) | 46 | 84 | 1 | 0 | 2 |
| Sodium Chloride (NaCl) | 0 | 1 | 0 | 0 | 0 |
| Strontium (Sr) | 10 | 0 | 0 | 0 | 0 |
| Sulfate (SO$_4^{2-}$) | 63 | 102 | 4 | 1 | 1 |
| Sulfide (S$^{2-}$) | 5 | 3 | 0 | 0 | 0 |
| Tellurium (Te) | 2 | 0 | 0 | 0 | 1 |
| Thallium (Tl) | 8 | 0 | 0 | 0 | 1 |
| Tin (Sn) | 2 | 0 | 0 | 0 | 1 |
| Titanium (Ti) | 1 | 0 | 0 | 0 | 1 |
| Tungsten (W) | 2 | 0 | 0 | 0 | 1 |
| Uranium (U) | 40 | 0 | 0 | 0 | 1 |
| Vanadium (V) | 9 | 1 | 0 | 0 | 1 |
| Zinc (Zn) | 60 | 72 | 0 | 1 | 2 |

[a] Six countries provide regulations for P$_2$O$_5$ rather than PO$_4^{3-}$. For the sake of comparison, the P$_2$O$_5$ regulations were converted to PO$_4^{3-}$ equivalent values.

In this study, we compared the standards from countries that do and do not distinguish between "mandatory", "indicator", or "no alternative" values by pooling these three categories into one group for summary statistics and statistical analyses since these values all apply to long-term exposures. The "emergency" values apply to short-term exposures, so they were not pooled with the "mandatory", "indicator", or "no alternative" values. Nevertheless, in our graphical depiction of the data in Figs 7, 11, and 15, we use differing colors to maintain the distinction between the different categories of regulations.

## National drinking water standards compared to World Health Organization Guideline values

Table 4 presents a comparison of national drinking water standards to the 2022 WHO GVs for each of the inorganic contaminants for which the WHO provides a GV. The mode, median, minimum, and maximum regulatory values, and the number of countries that have a regulation for a given contaminant are listed in this table. The mode and median values are based on the entire set of regulations, while the minimum and maximum exclude regulatory values for "emergencies". In addition, this table also shows the percentage of countries that meet, do not meet, or do not have a standard for each WHO GV; the percentage of the world's population living in countries in each of these categories is also listed. The income differences between each of these categories are calculated by summing the total population of the countries in each category and dividing by the total GDP of these countries (Population-Weighted or P-W GDP/capita). This summary of national regulation values includes and makes no distinction between "mandatory", "indicator", and "no alternative" values.

Fig 7 presents a visual comparison of the 2022 WHO GVs for inorganic contaminants to the current national drinking water standards. In this figure, "mandatory", "indicator", "no alternative", and "emergency" values are shown in grey, blue, orange, and pink, respectively. This figure highlights the fact that for each of the contaminant, there is at least one national regulation for long-term/chronic exposures that exceeds (less protective than) the associated WHO GV, but there are also national regulations below (more protective than) the associated WHO GV for all contaminants (Table 4 and Fig 7).

**Table 4. A comparison of the 2022 World Health Organization (WHO) Guideline Values (GVs) in milligrams/liter (mg/L) for inorganic contaminants in drinking water to the current national drinking water standards [18].** Minima and maxima are listed twice, first with all values as written in sources and then with potential typographical errors removed. Country categories are compared by raw count of countries and then by percentage of world population within each category. Income differences between categories are compared using population-weighted gross domestic product (P-W GDP)/capita within each category, shown in United States dollars (USD).

| Contaminant | WHO GV (mg/L) | Mode of all regulation values (mg/L) | Median of all regulation values (mg/L) | Minimum of long-term regulation values (mg/L)[a] (all data; typos removed[b]) | Maximum of long-term regulation values (mg/L)[a] (all data; typos removed[b]) | Countries with regulations (count; % population[c]; P-W GDP/capita in USD[d]) | Countries with regulations less than or equal to WHO GV (count; % population[c]; P-W GDP/capita in USD[d]) | Countries with regulations greater (less protective) than WHO GV (count; % population[c]; P-W GDP/capita in USD[d]) | Countries with no regulation (count; % population[c]; P-W GDP/capita in USD[d]) |
|---|---|---|---|---|---|---|---|---|---|
| Antimony (Sb) | 0.02 | 0.005 | 0.005 | 0.003 | 0.1 | 115 | 113 | 2 | 80 |
| | | | | 0.003 | 0.1 | 56% | 56% | 0.2% | 44% |
| | | | | | | $15,667 | $15,694 | $5,946 | $4,841 |
| Arsenic (As) | 0.01 | 0.01 | 0.01 | 0.001 | 0.5 | 175 | 131 | 44 | 20 |
| | | | | 0.005 | 0.3 | 98% | 66% | 32% | 2% |
| | | | | | | $11,111 | $15,230 | $2,707 | $1,039 |
| Barium (Ba) | 1.3 | 0.7 | 0.7 | 0.01 | 2 | 104 | 99 | 5 | 91 |
| | | | | 0.1 | 2 | 83% | 77% | 6% | 17% |
| | | | | | | $9,005 | $5,810 | $52,185 | $21,022 |
| Boron (B) | 2.4 | 1 | 1 | 0.001 | 5 | 127 | 124 | 3 | 68 |
| | | | | 0.1 | 5 | 79% | 78% | 1% | 21% |
| | | | | | | $9,370 | $8,989 | $39,405 | $17,015 |
| Cadmium (Cd) | 0.003 | 0.003 | 0.003 | 0.001 | 0.02 | 167 | 95 | 72 | 28 |
| | | | | 0.001 | 0.02 | 97% | 57% | 41% | 3% |
| | | | | | | $11,162 | $5,949 | $18,589 | $1,280 |
| Chromium (Cr)[e] | 0.05 | 0.05 | 0.05 | 0.01 | 0.2 | 164 | 160 | 4 | 31 |
| | | | | 0.01 | 0.2 | 96% | 92% | 4% | 4% |
| | | | | | | $11,189 | $8,737 | $62,940 | $3,378 |
| Copper (Cu) | 2 | 2 | 2 | .002 | 1500 | 170 | 166 | 4 | 25 |
| | | | | | | 97% | 97% | 1% | 3% |
| | | | | | | $11,123 | $11,150 | $3,988 | $2,012 |
| Fluoride (F⁻) | 1.5 | 1.5 | 1.5 | 0.01 | 5 | 162 | 154 | 8 | 33 |
| | | | | 0.6 | 5 | 94% | 89% | 5% | 6% |
| | | | | | | $11,398 | $8,873 | $53,266 | $2,945 |
| Lead (Pb) | 0.01 | 0.01 | 0.01 | 0.005 | 10 | 172 | 131 | 41 | 23 |
| | | | | 0.005 | 0.1 | 97% | 80% | 17% | 3% |
| | | | | | | $11,126 | $9,307 | $20,097 | $2,184 |
| Manganese (Mn) | 0.08 | 0.05 | 0.1 | 0.02 | 1 | 163 | 70 | 93 | 32 |
| | | | | 0.02 | 1 | 97% | 18% | 79% | 3% |
| | | | | | | $11,156 | $34,826 | $5,737 | $2,243 |
| Mercury (Hg) | 0.006 | 0.001 | 0.001 | 0.0001 | 0.01 | 166 | 163 | 3 | 29 |
| | | | | 0.0001 | 0.01 | 78% | 77% | 1% | 22% |
| | | | | | | $11,484 | $11,511 | $8,111 | $8,962 |
| Nickle (Ni) | 0.07 | 0.02 | 0.02 | 0.01 | 0.5 | 141 | 130 | 11 | 54 |
| | | | | 0.01 | 0.5 | 85% | 79% | 5% | 15% |
| | | | | | | $8,043 | $8,181 | $5,995 | $27,597 |
| Nitrate (NO₃⁻) | 50 | 50 | 50 | 0 | 70 | 175 | 172[f] | 1[f] | 20 |
| | | | | 0 | 70 | 96% | 96% | 0.1% | 4% |
| | | | | | | $10,596 | $10,561 | $44,169 | $20,601 |

*(Continued)*

**Table 4.** (Continued)

| Contaminant | WHO GV (mg/L) | Mode of all regulation values (mg/L) | Median of all regulation values (mg/L) | Minimum of long-term regulation values (mg/L)[a] (all data; typos removed[b]) | Maximum of long-term regulation values (mg/L)[a] (all data; typos removed[b]) | Countries with regulations (count; % population[c]; P-W GDP/capita in USD[d]) | Countries with regulations less than or equal to WHO GV (count; % population[c]; P-W GDP/capita in USD[d]) | Countries with regulations greater (less protective) than WHO GV (count; % population[c]; P-W GDP/capita in USD[d]) | Countries with no regulation (count; % population[c]; P-W GDP/capita in USD[d]) |
|---|---|---|---|---|---|---|---|---|---|
| Nitrite (NO$_2^-$) | 3 | 3 | 0.5 | 0 | 3.3 | 147 | 135[f] | 9[f] | 48 |
| | | | | 0 | 3.3 | 55% | 47% | 7% | 45% |
| | | | | | | | $14,527 | $10,647 | $38,620 | $6,587 |
| Selenium (Se) | 0.04 | 0.01 | 0.01 | 0.01 | 0.1 | 160 | 153 | 7 | 35 |
| | | | | 0.01 | 0.05 | 93% | 85% | 8% | 7% |
| | | | | | | $11,546 | $9,047 | $36,628 | $1,834 |
| Uranium (U) | 0.03 | 0.03 | 0.03 | 0.002 | 4 | 40 | 39 | 1 | 155 |
| | | | | 0.002 | 4 | 19% | 19% | 0.03% | 81% |
| | | | | | | $23,563 | $23,597 | $4,179 | $7,942 |

Abbreviations: GV: Guideline value; mg/L = milligrams/liter; NA = Not available; P-W GDP = Population-Weighted Gross Domestic Product; WHO = World Health Organization, USD = United States Dollars.

[a] Long-term regulation values include "mandatory", "indicator", and "no alternative" values, which all apply to long-term exposures, but not "emergency" values, which are generally understood only to involve exposures of limited duration.

[b] As described in Section 3.2.4, national regulatory limits that contained the same numerals as the WHO GV or other common regulatory values for the same contaminant but differed in the placement of the decimal point were identified as potential typographical errors. In this table, minima and maxima are calculated twice. The first value is with potential errors included ("all data"), and the second value is with the potential errors removed ("typos removed").

[c] In this table, "% population" refers to the percentage of total world population that falls into each category. These values were calculated by summing the populations of the individual countries that fall into each category, then dividing by the total population of all 195 countries. Population data for 2020 were drawn from World Bank [25].

[d] In this table, population-weighted gross domestic products per capita (P-W GDPs/capita) were calculated by summing the GDPs for all countries that fall into each category (for which GDP data were available), then dividing this number by the sum of the total population for all countries that fall into this category (for which GDP data were available). GDP data for 2020 were drawn from the World Bank [26].

[e] The WHO drinking-water guideline is specified for total Cr. Two countries provide regulations for Cr(III) with a maximum of 0.5 mg/L as well as a maximum of 0.05 mg/L for Cr(VI). Two countries provide regulations for Cr(VI) of 0.05 mg/L as well as total Cr of 0.05 mg/L; one country provides a regulation for Cr(VI) of 0.05 mg/L as well as a regulation for total Cr of 0.01 mg/L [sic]. Twelve countries provide a regulation only for Cr(VI). Thirty-three countries provide a regulation specifically for total Cr. The regulations for the remaining countries do not state whether they apply to total Cr or Cr(VI).

[f] Two countries have regulatory maximum levels for NO$_3^-$ concentration plus NO$_2^-$ concentration but do not provide specific maximum levels for NO$_3^-$ concentration or NO$_2^-$ concentration separately, so these countries were omitted from tallies of countries that did or did not meet WHO GVs for these contaminants.

For most contaminants for which there are WHO GVs, more than 90% of the world's population live in countries where there are national standards for these contaminants. However, for U, only 20% of the world's population lives in countries that have national standards (Table 4 and Fig 8). Substantial populations live in countries that have drinking water regulations but whose regulations exceed (are less protective than) WHO GVs for As, Cd, Mn, and Pb (Table 4 and Fig 8).

For most contaminants with WHO GVs, P-W GDPs/capita (total GDPs/total populations) are substantially higher for populations living in countries with regulations as compared to populations living in countries without regulations for these contaminants (Table 4 and Fig 9). However, for B, Ba, Ni, and NO$_3^-$, the P-W GDPs/capita are higher for populations living in countries without regulations for these contaminants than for populations living in countries with regulations. (Table 4 and Fig 9).

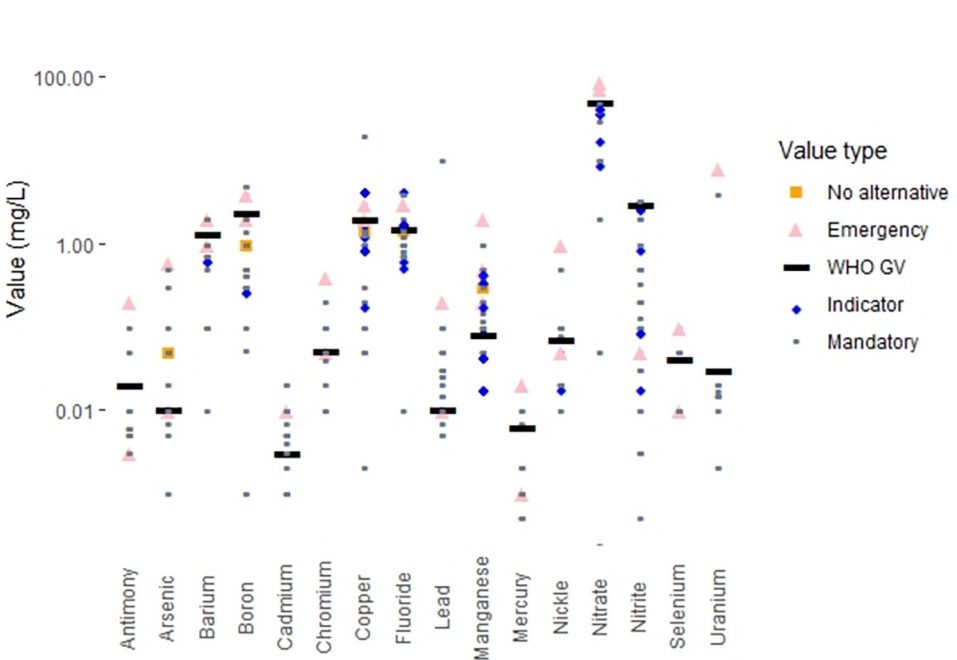

**Fig 7. A visual comparison of the 2022 World Health Organization (WHO) Guideline Values (GVs) for inorganic contaminants in drinking water to the current national drinking water standards [18].** Values are shown in milligrams/liter (mg/L) on a logarithmic scale.

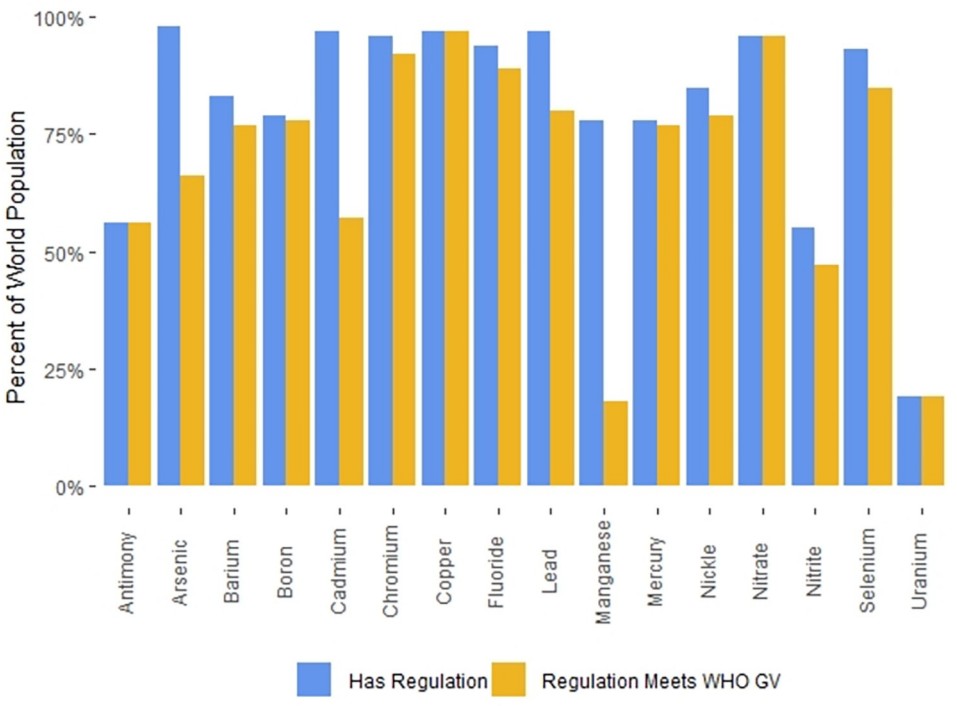

**Fig 8. Percentage of 2020 world population living in countries that have regulations and whose regulations meet World Health Organization (WHO) guideline values (GVs).** Population data from the World Bank [25].

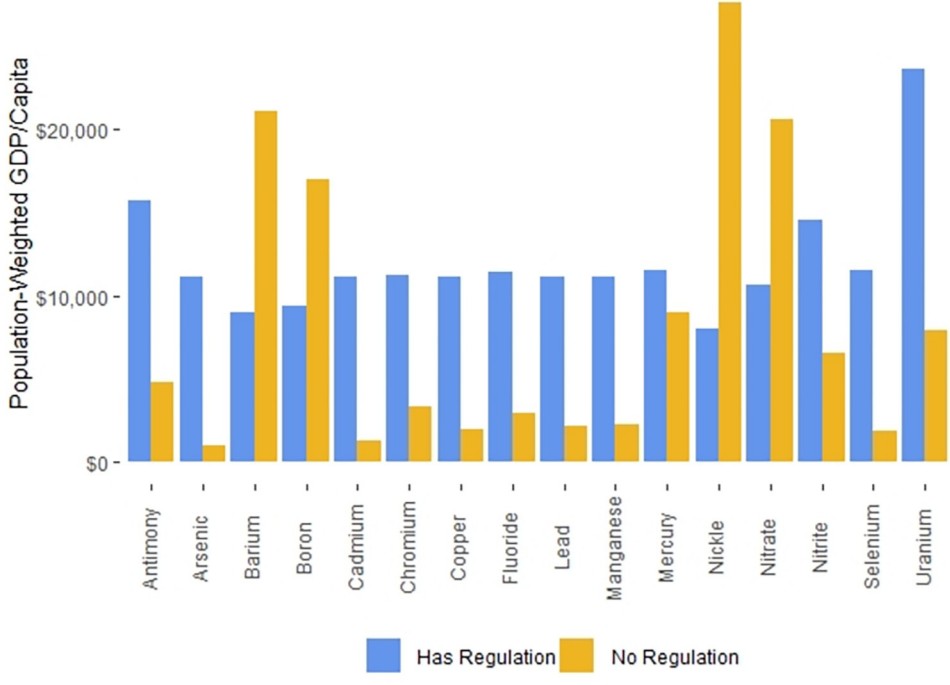

**Fig 9. Population-weighted gross domestic products (P-W GDPs) /capita in United States dollars (USD) for populations living in countries that have or do not have regulations for contaminants with World Health Organization (WHO) guideline values (GVs).** Population and GDP data are from the World Bank for 2020 [25, 26].

P-W GDPs/capita are substantially higher for populations living in countries whose regulations for, As, Cu, Hg, Mn, Sb, and U meet the WHO GVs for these contaminants than for populations living in countries whose regulations exceed (are less protective than) WHO GVs (Fig 10). In contrast, P-W GDPs/capita are substantially lower for populations living in countries whose regulations for B, Ba, Cd, Cr, F⁻, NO₃⁻, NO₂⁻, Pb and Se meet the WHO GVs for these contaminants than for populations living in countries whose regulations exceed (are less protective than) the WHO GVs.

### National drinking water standards compared to World Health Organization Health-Based Values and Reference Values

Table 5 compares national drinking water standards for contaminants for which the WHO currently has HBVs or RVs but no formal GVs. The mode, median, minimum, and maximum regulatory values and the number of countries that have a regulation for a given contaminant are listed in this table. The mode and median values are based on the entire set of regulations; however, the minimum and maximum columns exclude regulatory values for "emergencies". In addition, this table also shows the percentage of countries that either meet, do not meet, or do not have a standard for each WHO HBV or health-based RV; the percentage of the world's population living in countries in each of these categories is also listed. The income differences between each of these categories are calculated by summing the total population of the countries in each category and dividing by the total GDP of these countries (population-weighted or P-W GDP/capita). This summary of national regulation values includes and makes no distinction between "mandatory", "indicator", and "no alternative" values.

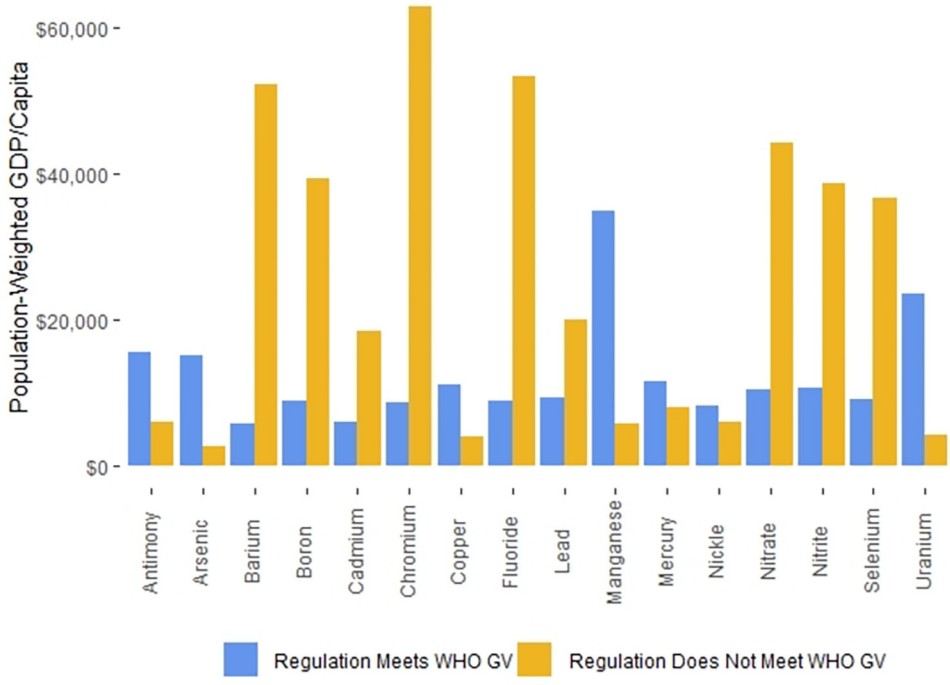

**Fig 10. Population-weighted Gross Domestic Products (P-W GGDPs) /capita in United States Dollars (USD) for populations living in countries whose regulations meet or exceed World Health Organization (WHO) guideline values (GVs) for inorganic contaminants.** Population and GDP data from the World Bank for 2020 [25, 26].

Fig 11 presents a visual comparison of the national drinking water standards to the 2022 WHO HBVs or RVs for contaminants for which the WHO provides HBVs or RVs but no formal GVs. In this figure, "mandatory", "indicator", "no alternative", and "emergency" values are shown in grey, blue, orange, and pink, respectively. This figure highlights that for all contaminants with HBVs, there are countries with national regulations below (more protective than) the HBVs. However, for Al, CN⁻, Fe, and Mo, there are also national regulations for chronic exposures that exceed (are less protective than) the WHO GVs (Fig 11).

For Al, CN⁻, and Fe, more than 90% of the world's population live in countries where there are national standards for these contaminants. However, less than 50% of the world's population live in countries for which there is a national drinking water regulation for Be (Table 5 and Fig 12).

P-W GDPs/capita are substantially higher for populations living in countries with regulations for Ag, Al, Be, CN⁻, and Fe (Table 4 and Fig 13). However, the P-W GDP/capita is higher for populations living in countries without a regulation for Mo than for populations living in countries with a regulation for Mo (Table 4 and Fig 13).

P-W GDPs/capita are substantially higher for populations living in countries with regulations that exceed (are less protective than) the WHO HBVs for Al and CN⁻ than for populations living in countries with regulations that meet the WHO HBVs for these contaminants. However, for Fe, the P-W GDP/capita is higher for populations living in countries with regulations that meet the WHO HBV than for populations whose national regulations exceed (are less protective than) the WHO HBV (Table 4 and Fig 14).

**Table 5. A comparison of national drinking water standards for inorganic contaminants for which the World Health Organization (WHO) has calculated Health-based Values (HBVs) or health-based Reference Values (RVs) in milligrams/liter (mg/L) but does not currently have formal Guideline Values (GVs) [18].** Minima and maxima are listed twice, first with all values as written in sources and then with potential typographical errors removed. Country categories are compared by raw count of countries and then by percentage of world population within each category. Income differences between categories are compared using population-weighted gross domestic projects (P-W GDP)/capita within each category, shown in United States dollars (USD).

| Contaminant | WHO HBV or health-based RV[a] (mg/L) | Mode of all regulation values (mg/L) | Median of all regulation values (mg/L) | Minimum of long-term regulation values (mg/L)[a] (all data; typos removed[b]) | Maximum of long-term regulation values (mg/L)[a] (all data; typos removed[b]) | Countries with regulations (count; % population[c]; GDP/capita USD[d]) | Countries with regulations less than or equal to WHO HBV or health-based RV (count; % population[c]; P-W GDP/capita in USD[d]) | Countries with regulations greater (less protective) than WHO HBV or health-based RV (count; % population[c]; P-W GDP/capita in USD[d]) | Countries with no regulation (count; % population[c]; P-W GDP/capita in USD[d]) |
|---|---|---|---|---|---|---|---|---|---|
| Aluminum (Al) | 0.9 | 0.2 | 0.2 | 0.0002 | 200 | 146 | 145 | 1 | 49 |
| | | | | 0.03 | 0.9 | 94% | 94% | 0.1% | 6% |
| | | | | | | $11,238 | $11,230 | $24,812 | $5,361 |
| Beryllium (Be) | 0.012 | 0.0002 | 0.003 | 0 | 0.12 | 20 | 18 | 2 | 175 |
| | | | | 0 | 0.12 | 28% | 27% | 1% | 72% |
| | | | | | | $18,429 | $18,419 | $18,612 | $7,860 |
| Cyanide (CN⁻) | 0.07 | 0.05 | 0.05 | 0.001 | 0.6 | 140 | 139 | 1 | 55 |
| | | | | 0.001 | 0.6 | 91% | 91% | 0.1% | 9% |
| | | | | | | $11,563 | $11,541 | $41,441 | $3,961 |
| Iron (Fe) | 2 | 0.3 | 0.3 | 0 | 3 | 165 | 164 | 1 | 30 |
| | | | | 0.2 | 3 | 95% | 94% | 0.4% | 5% |
| | | | | | | $11,358 | $11,399 | $1,155 | $2,852 |
| Molybdenum (Mo) | 0.07 | 0.07 | 0.07 | 0.01 | 0.25 | 44 | 35 | 9 | 151 |
| | | | | 0.01 | 0.25 | 53% | 50% | 3% | 47% |
| | | | | | | $6,347 | $6,254 | $7,838 | $16,117 |
| Silver (Ag) | 0.1[e] | 0.05, 0.1 | 0.05 | 0.001 | 0.1 | 41 | 41 | 0 | 154 |
| | | | | 0.001 | 0.1 | 50% | 50% | 0% | 50% |
| | | | | | | $11,802 | $11,802 | NA | $10,046 |

Abbreviations: GV: Guideline value; mg/L = milligrams/liter; NA = Not available; P-W GDP = Population-Weighted Gross Domestic Product; WHO = World Health Organization, USD = United States Dollars.

[a] Long-term regulation values include "mandatory", "indicator", and "no alternative" values, which all apply to long-term exposures, but not "emergency" values, which are generally understood only to involve exposures of limited duration.

[b] As described in Section 3.2.4, national regulatory limits that contained the same numerals as the WHO GV or other common regulatory values for the same contaminant but differed in the placement of the decimal point were identified as apparent typographical errors. In this table, minima and maxima are calculated twice. The first value is with potential errors included ("all data"), and the second value is with the apparent errors removed ("typos removed").

[c] In this table, "% population" refers to the percentage of total world population that falls into each category. These values were calculated by summing the populations of the individual countries that fall into each category, then dividing by the total population of all 195 countries. Population data for 2020 were drawn from the World Bank [25].

[d] In this table, population-weighted gross domestic products per capita (P-W GDPs/capita) were calculated by summing the GDPs for all countries that fall into each category (for which GDP data were available), then dividing this number by the sum of the total population for all countries that fall into this category (for which GDP data were available). GDP data for 2020 were drawn from the World Bank [26].

[e] This is termed a health-based "reference value" (RV) or a "bounding value" by the WHO rather than a formal "health-based value" (HBV) [18, 32]. The distinction between a health-based "reference value" or a "bounding value" and a formal "health-based value" is not entirely clear [18, 32].

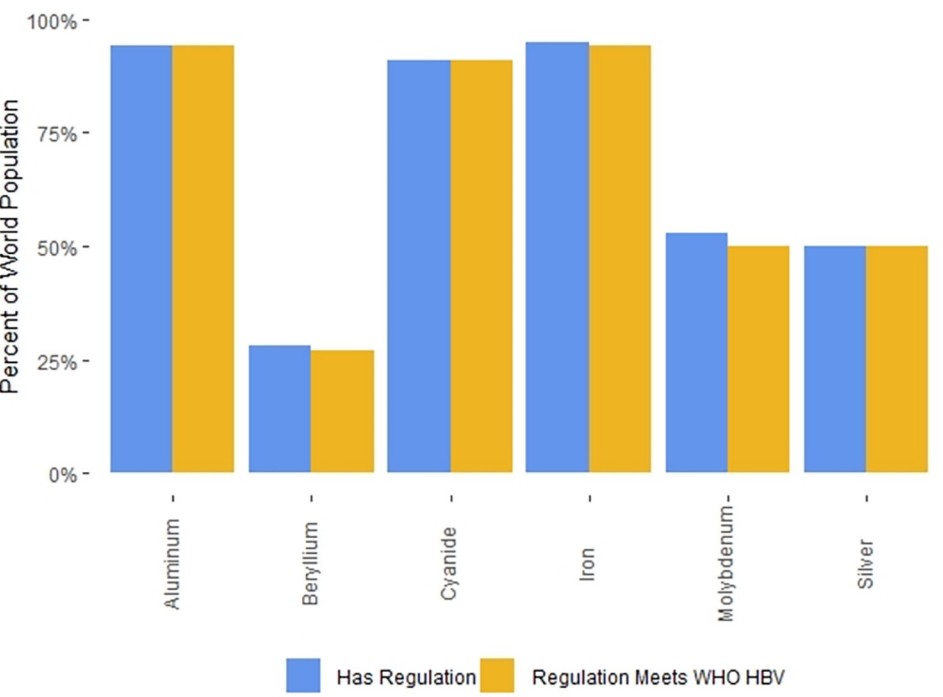

**Fig 11. A visual comparison of national drinking water standards for inorganic contaminants for which the World Health Organization (WHO) provides a health-based value (HBV) but no drinking water guideline value (GV).** Values are shown in milligrams/liter (mg/L) on a logarithmic scale.

**Fig 12. Percentage of world population living in countries that have regulations and whose regulations meet World Health Organization (WHO) health-based values (HBVs) or reference values (RVs).** Population data are from the World Bank for 2020 [25].

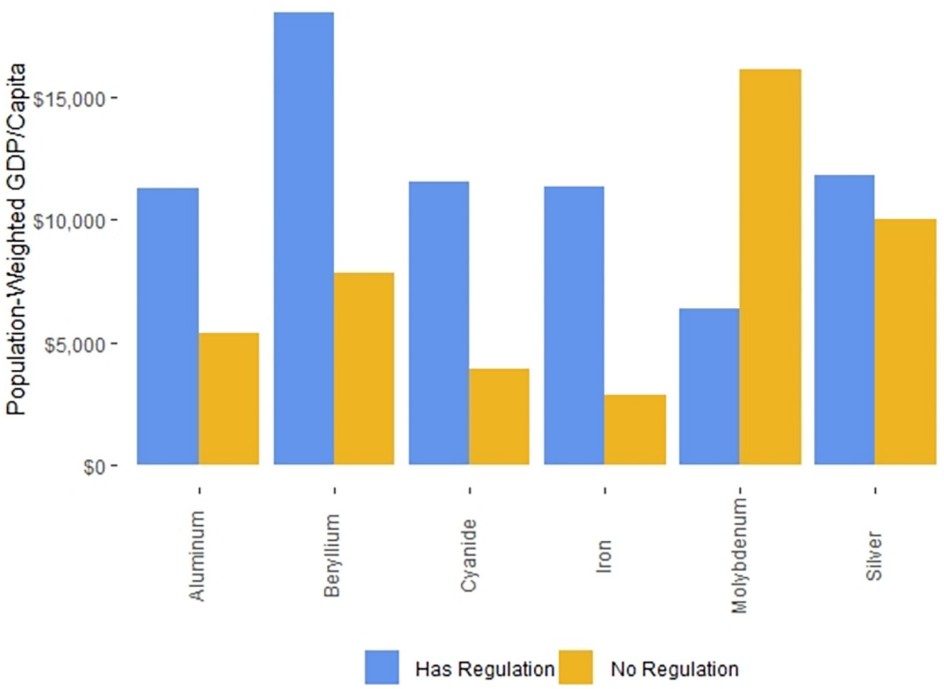

**Fig 13. Population-weighted Gross Domestic Products (P-W GDPs)/capita in United States dollars (USD) for populations living in countries that have or do not have regulations for contaminants with World Health Organization (WHO) health-based values (HBVs) or Reference Values (RVs).** Population and GDP data from the World Bank for 2020 [25, 26].

## National drinking water standards compared to World Health Organization Aesthetic Values and national drinking water standards for contaminants with no World Health Organization guidance

Table 6 presents a list of national drinking water standards for inorganic contaminants for which the WHO has not calculated a GV, HBV, or health-based RV; the WHO has provided an AV for some of these contaminants. The mode, median, minimum, and maximum regulatory values, and the number of countries that have such a regulation for a given contaminant are listed in this table. The mode and median values are based on the entire set of regulations; in contrast, the minimum and maximum do not include regulatory values for "emergencies". In addition, this table also shows the percentage of countries that either meet, do not meet, or do not have a standard for each WHO AV; the percentage of the world's population living in countries in each of these categories is also listed. The income differences between each of these categories are calculated by summing the total population of the countries in each category and dividing by the total GDP of these countries (population-weighted or P-W GDP/capita). This summary of national regulation values includes and makes no distinction between "mandatory", "indicator", and "no alternative" values.

Fig 15 presents a visual comparison of the inorganic contaminants that do not have 2022 WHO GVs, HBVs, or health-based RVs [18] and appear in five or more national drinking water standards. In this figure, "mandatory", "indicator", "no alternative", and "emergency" values are shown in grey, blue, orange, and pink, respectively. Some of these contaminants have WHO AVs. This figure highlights the fact that for four of the six contaminants for which the WHO provides an AV, Cl⁻, NH₃, Na, and Zn, there are national regulations which exceed

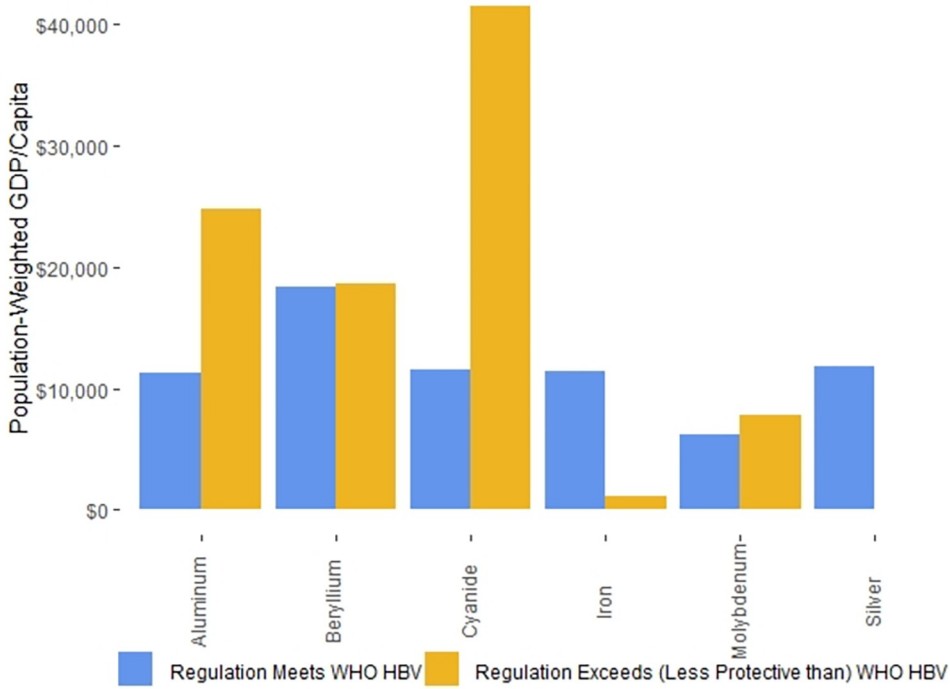

**Fig 14. Population-weighted Gross Domestic Products (P-W GDPs)/capita in United States dollars (USD) for populations living in countries whose regulations meet or exceed (less protective than) World Health Organization (WHO) health-based values (HBVs) or health-based reference values (RVs) for inorganic contaminants.** Population and GDP data are from the World Bank for 2020 [25, 26].

(are less protective than) this AV (Fig 15). At the same time, for all of these contaminants with WHO AVs, many national regulations have values that are much lower than the WHO AVs.

More than 50% of the world's population live in countries where there are national standards for $Cl^-$, Na, $SO_4^{2-}$, and Zn. However, less than 50% of the world's population live in countries for which there is a national drinking water regulation for HS⁻ or $NH_3$ (Table 5 and Fig 16).

For HS⁻, $NH_3$, and Zn, P-W GDPs/capita are substantially higher for populations living in countries without regulations than for populations living in countries with regulations (Table 7 and Fig 17).

In contrast, with the exceptions of HS⁻ and $NH_3$, P-W GDPs/capita are substantially higher for populations living in countries with regulations that meet the WHO AVs compared to those living in countries exceed (are less protective than) the WHO AVs. (Table 7 and Fig 18).

Fig 19 highlights the fact that Mg, $S^{2-}$, Ca, N, and Tl all appear in regulations that apply to at least 20% of the world's population, although there is no WHO guidance for these contaminants in drinking water.

Fig 20 highlights the fact that, when focusing on those inorganic contaminants that are regulated by five or more countries, P-W GDPs/capita are higher than $10,000 USD for populations living in countries with drinking water regulations forTl, $NH_4^+$, V, Sr, and $O_3$, but lower than $5,000 USD for populations living in countries with drinking water regulations for Mg, Ca, K, $PO_4^{3-}$, and P.

**Table 6. A list of current national drinking water standards in milligrams/liter (mg/L) for inorganic contaminants that do not have 2022 World Health Organization (WHO) guideline values (GVs), health-based values (HBVs) or health-based reference values (RVs) [18].** Minima and maxima are listed twice, first with all values as written in sources and then with potential typographical errors removed. Country categories are compared by raw count of countries and then by percentage of world population within each category. Income differences between categories are compared using population-weighted gross domestic projects (P-W GDP)/capita within each category, shown in United States dollars (USD).

| Contaminant | WHO AV (mg/L) | Mode of all regulation values (mg/L) | Median of all regulation values (mg/L) | Minimum of long-term regulation values (mg/L)[a] (all data; typos removed[b]) | Maximum of long-term regulation values (mg/L)[a] | Countries with regulations (count; % population[c]; GDP/capita USD[c]) | Countries with regulations less than or equal to WHO AV (count; % population[c]; P-W GDP/capita in USD[d]) | Countries with regulations greater (less protective) than WHO AV (count; % population[c]; P-W GDP/capita in USD[d]) | Countries with no regulation (count; % population[c]; P-W GDP/capita in USD[c]) |
|---|---|---|---|---|---|---|---|---|---|
| Ammonia ($NH_3$) | 1.5 | 1.5 | 1.5 | 0 | 3 | 65 | 56[e] | 8[d] | 130 |
| | | | | 0 | 3 | 45% | 40% | 3% | 55% |
| | | | | | | $3,814 | $3,816 | $5,181 | $16,757 |
| Ammonium ($NH_4^+$) | NA | 0.5 | 0.5 | 0 | 2.6 | 64 | NA | NA | 131 |
| | | | | 0 | 2.6 | 15% | | | 85% |
| | | | | | | $20,073 | | | $9,349 |
| Bicarbonate ($HCO_3^-$) | NA | 500 | 500 | 500 | 500 | 1 | NA | NA | 19 |
| | | | | 500 | 500 | 0.6% | | | 99.4% |
| | | | | | | $486 | | | $10,989 |
| Bismuth (Bi) | NA | 0.1, 0.5, 1.0[e] | 0.5 | 0.1 | 0.5 | 2 | NA | NA | 193 |
| | | | | 0.1 | 0.5 | 0.2% | | | 99.8% |
| | | | | | | $5,946 | | | $10,936 |
| Borate ($BO_3^{3-}$) | NA | 0.2 | 0.2 | 0.2 | 0.2 | 1 | NA | NA | 194 |
| | | | | 0.2 | 0.2 | 0.01% | | | 99.99% |
| | | | | | | $25,194 | | | $10,927 |
| Calcium (Ca) | NA | 200 | 150 | 25 | 300 | 52 | NA | NA | 143 |
| | | | | 25 | 300 | 36% | | | 64% |
| | | | | | | $2,661 | | | $15,484 |
| Cerium (Ce) | NA | 2, 4[e] | 3[e] | 2 | 2 | 1 | NA | NA | 194 |
| | | | | 2 | 2 | 0.03% | | | 99.97% |
| | | | | | | $4,179 | | | $10,930 |
| Chloride ($Cl^-$) | 300 | 250 | 250 | 0.7 | 1000 | 163 | 131 | 32 | 32 |
| | | | | 50 | 1000 | 95% | 65% | 30% | 5% |
| | | | | | | $11,063 | $14,492 | $3,412 | $8,008 |
| Cobalt (Co) | NA | 0.5 | 0.5 | 0.002 | 12 | 13 | NA | NA | 182 |
| | | | | 0.002 | 12 | 4% | | | 96% |
| | | | | | | $7,328 | | | $11,067 |
| Europium (Eu) | NA | 0.3 | 0.3 | 0.3 | 0.3 | 1 | NA | NA | 194 |
| | | | | 0.3 | 0.3 | 0.1% | | | 99.9% |
| | | | | | | $6,424 | | | $10,934 |
| Fluorine (F) | NA | 0.02 | 0.02 | 0.02 | 0.02 | 1 | NA | NA | 194 |
| | | | | 0.02 | 0.02 | 0.5% | | | 99.5% |
| | | | | | | $4,146 | | | $10,964 |
| Gold (Au) | NA | 0.005, 0.01[e] | 0.0075[e] | 0.005 | 0.005 | 1 | NA | NA | 194 |
| | | | | 0.005 | 0.005 | 0.03% | | | 99.97% |
| | | | | | | $4,179 | | | $10,930 |
| Hydrogen sulfide ($H_2S$) | 1 | 0.1 | 0.075 | 0 | 1 | 44 | 43 | 1 | 151 |
| | | | | 0 | 1 | 12.08% | 12.05% | 0.03% | 87.92% |
| | | | | | | $4,493 | $4,494 | $4,179 | $11,832 |

(*Continued*)

**Table 6.** (Continued)

| Contaminant | WHO AV (mg/L) | Mode of all regulation values (mg/L) | Median of all regulation values (mg/L) | Minimum of long-term regulation values (mg/L)[a] (all data; typos removed[b]) | Maximum of long-term regulation values (mg/L)[a] | Countries with regulations (count; % population[c]; GDP/capita USD[c]) | Countries with regulations less than or equal to WHO AV (count; % population[c]; P-W GDP/capita in USD[d]) | Countries with regulations greater (less protective) than WHO AV (count; % population[c]; P-W GDP/capita in USD[d]) | Countries with no regulation (count; % population[c]; P-W GDP/capita in USD[c]) |
|---|---|---|---|---|---|---|---|---|---|
| Hydrosulfide (HS⁻) | NA | 3 | 3 | 3 | 3 | 2 | NA | NA | 193 |
| | | | | 3 | 3 | 0.3% | | | 99.7% |
| | | | | | | $5,282 | | | $10,942 |
| Indium (In) | NA | 0.07 | 0.07 | 0.07 | 0.07 | 1 | NA | NA | 194 |
| | | | | 0.07 | 0.07 | 0.3% | | | 99.7% |
| | | | | | | NA | | | $10,928 |
| Iodide (I⁻) | NA | 0.5, 1, 2[e] | 1 | 0.5 | 1 | 2 | NA | NA | 193 |
| | | | | 0.5 | 1 | 0.4% | | | 99.6% |
| | | | | | | $47,416 | | | $10,792 |
| Iodine (I) | NA | 500 | 500 | 500 | 500 | 1 | NA | NA | 193 |
| | | | | 500 | 500 | 2% | | | 98% |
| | | | | | | $1,962 | | | $11,127 |
| Lanthanum (La) | NA | 0.002 | 0.002 | 0.002 | 0.002 | 1 | NA | NA | 194 |
| | | | | 0.002 | 0.002 | 0.3% | | | 99.7% |
| | | | | | | $51,693 | | | $10,790 |
| Lithium (Li) | NA | 0.03, 0.05, 5, 10[e] | 2.525 | 0.03 | 5 | 3 | NA | NA | 192 |
| | | | | 0.03 | 5 | 0.2% | | | 99.8% |
| | | | | | | $14,545 | | | $10,921 |
| Magnesium (Mg) | NA | 50 | 85 | 10 | 500 | 63 | NA | NA | 132 |
| | | | | 10 | 500 | 39% | | | 61% |
| | | | | | | $3,099 | | | $15,960 |
| Niobium (Nb) | NA | 0.01 | 0.01 | 0.01 | 0.01 | 1 | NA | NA | 194 |
| | | | | 0.01 | 0.01 | 0.1% | | | 99.9% |
| | | | | | | $6,424 | | | $10,934 |
| Nitrogen (N) | NA | 1 | 1 | 0.5 | 11 | 10 | NA | NA | 185 |
| | | | | 0.5 | 11 | 24% | | | 76% |
| | | | | | | $8,866 | | | $11,607 |
| Dissolved oxygen (O₂) | NA | NA[f] | NA[f] | NA[f] | NA[f] | 15 | NA | NA | 180 |
| | | | | | | 5% | | | 95% |
| | | | | | | $8,556 | | | $11,059 |
| Ozone (O₃) | NA | 0.05 | 0.05 | 0.05 | 0.3 | 5 | NA | NA | 190 |
| | | | | 0.05 | 0.3 | 1% | | | 99% |
| | | | | | | $16,271 | | | $10,851 |
| Phosphate (PO₄³⁻)[g] | NA | 0.5, 2.2 | 1.05 | 0.015 | 6 | 24 | NA | NA | 171 |
| | | | | 0.015 | 6 | 8% | | | 92% |
| | | | | | | $4,777 | | | $11,462 |
| Phosphorus (P) | NA | 2.2 | 0.44 | 0 | 5 | 15 | NA | NA | 180 |
| | | | | 0 | 5 | 5% | | | 95% |
| | | | | | | $3,856 | | | $11,268 |
| Potassium (K) | NA | 12 | 12 | 1.5 | 400 | 29 | NA | NA | 166 |
| | | | | 1.5 | 400 | 9% | | | 91% |
| | | | | | | $4,030 | | | $11,590 |

(*Continued*)

**Table 6.** (Continued)

| Contaminant | WHO AV (mg/L) | Mode of all regulation values (mg/L) | Median of all regulation values (mg/L) | Minimum of long-term regulation values (mg/L)[a] (all data; typos removed[b]) | Maximum of long-term regulation values (mg/L)[a] | Countries with regulations (count; % population[c]; GDP/capita USD[c]) | Countries with regulations less than or equal to WHO AV (count; % population[c]; P-W GDP/capita in USD[d]) | Countries with regulations greater (less protective) than WHO AV (count; % population[c]; P-W GDP/capita in USD[d]) | Countries with no regulation (count; % population[c]; P-W GDP/capita in USD[c]) |
|---|---|---|---|---|---|---|---|---|---|
| Rhodium (Rh) | NA | 0.1 | 0.1 | 0.1 | 0.1 | 1 | NA | NA | 194 |
| | | | | 0.1 | 0.1 | 0.1% | | | 99.9% |
| | | | | | | $6,424 | | | $10,934 |
| Rubidium (Rb) | NA | 0.1 | 0.1 | 0.1 | 0.1 | 1 | NA | NA | 194 |
| | | | | 0.1 | 0.1 | 0.1% | | | 99.9% |
| | | | | | | $6,424 | | | $10,934 |
| Samarium (Sm) | NA | 0.024 | 0.024 | 0.024 | 0.024 | 1 | NA | NA | 194 |
| | | | | 0.024 | 0.024 | 0.1% | | | 99.9% |
| | | | | | | $6,424 | | | $10,934 |
| Silicone dioxide ($SiO_2$) | NA | 80 | 80 | 80 | 80 | 1 | NA | NA | 194 |
| | | | | 80 | 80 | 0.3% | | | 99.7% |
| | | | | | | $51,693 | | | $10,790 |
| Sodium (Na) | 200 | 200 | 200 | 30 | 500 | 129 | 124 | 5 | 66 |
| | | | | 30 | 500 | 61.2% | 60.4% | 0.8% | 38.8% |
| | | | | | | $10,991 | $11,049 | $3,130 | $10,828 |
| Sodium Chloride (NaCl) | NA | 350 | 350 | 350 | 350 | 1 | NA | NA | 194 |
| | | | | 350 | 350 | 0.09% | | | 99.91% |
| | | | | | | $2,630 | | | $10,936 |
| Strontium (Sr) | NA | 7 | 7 | 4 | 7 | 10 | NA | NA | 185 |
| | | | | 4 | 7 | 4% | | | 96% |
| | | | | | | $11,889 | | | $10,886 |
| Sulfate ($SO_4^{2-}$) | 250 | 250 | 250 | 25 | 1000 | 161 | 113 | 48 | 34 |
| | | | | 25 | 1000 | 93% | 57% | 36% | 7% |
| | | | | | | $14,947 | $10,304 | $3,817 | $13,468 |
| Sulfide ($S^{2-}$) | NA | 0.05 | 0.035 | 0 | 0.05 | 8 | NA | NA | 187 |
| | | | | 0 | 0.05 | 39% | | | 61% |
| | | | | | | $6,621 | | | $13,824 |
| Tellurium (Te) | NA | 0.01 | 0.01 | 0.005 | 0.01 | 2 | NA | NA | 193 |
| | | | | 0.005 | 0.01 | 0.2% | | | 99.8% |
| | | | | | | $5,946 | | | $10,936 |
| Thallium (Tl) | NA | 0.002 | 0.002 | 0.0001 | 0.01 | 8 | NA | NA | 187 |
| | | | | 0.0001 | 0.01 | 23% | | | 77% |
| | | | | | | $20,480 | | | $8,036 |
| Tin (Sn) | NA | 0.2, 0.4, 2[e] | 0.4 | 0.2 | 2 | 2 | NA | NA | 193 |
| | | | | 0.2 | 2 | 2% | | | 98% |
| | | | | | | $1,995 | | | $11,129 |
| Titanium (Ti) | NA | 0.5, 1[e] | 0.75[e] | 0.5 | 0.5 | 1 | NA | NA | 194 |
| | | | | 0.5 | 0.5 | 0.03% | | | 99.97% |
| | | | | | | $4,179 | | | $10,930 |
| Tungsten (W) | NA | 0.05, 0.5, 1[e] | 0.5 | 0.05 | 0.5 | 2 | NA | NA | 193 |
| | | | | 0.05 | 0.5 | 0.2% | | | 99.8% |
| | | | | | | $5,946 | | | $10,936 |

(*Continued*)

**Table 6.** (Continued)

| Contaminant | WHO AV (mg/L) | Mode of all regulation values (mg/L) | Median of all regulation values (mg/L) | Minimum of long-term regulation values (mg/L)[a] (all data; typos removed[b]) | Maximum of long-term regulation values (mg/L)[a] | Countries with regulations (count; % population[c]; GDP/capita USD[c]) | Countries with regulations less than or equal to WHO AV (count; % population[c]; P-W GDP/capita in USD[d]) | Countries with regulations greater (less protective) than WHO AV (count; % population[c]; P-W GDP/capita in USD[d]) | Countries with no regulation (count; % population[c]; P-W GDP/capita in USD[c]) |
|---|---|---|---|---|---|---|---|---|---|
| Vanadium (V) | NA | 0.005, 0.1, 0.2 | 0.1 | 0 | 0.5 | 10 | NA | NA | 185 |
| | | | | 0 | 0.5 | 4% | | | 96% |
| | | | | | | $10,249 | | | $10,958 |
| Zinc (Zn) | 5 | 5 | 5 | 0.05 | 70 | 132 | 124 | 8 | 63 |
| | | | | 0.1 | 15 | 89% | 69% | 20% | 11% |
| | | | | | | $9,591 | $11,685 | $2,199 | $22,022 |

Abbreviations: GV: Guideline value; mg/L = milligrams/liter; NA = Not available; P-W GDP = Population-Weighted Gross Domestic Product; WHO = World Health Organization, USD = United States Dollars.

[a] Long-term regulation values include "mandatory", "indicator", and "no alternative" values, which all apply to long-term exposures, but not "emergency" values, which are generally understood only to involve shorter term exposures, which are of limited duration.

[b] As described in Section 3.2.4, national regulatory limits that contained the same numerals as the WHO GV or other common regulatory values for the same contaminant but differed in the placement of the decimal point were identified as potential typographical errors. In this table, minima and maxima are calculated twice. The first value is with potential errors included ("all data"), and the second value is with the potential errors removed ("typos removed").

[c] In this table, "% population" refers to the percentage of total world population that falls into each category. These values were calculated by summing the populations of the individual countries that fall into each category, then dividing by the total population of all 195 countries. Population data for 2020 were drawn from the World Bank [25].

[d] In this table, population-weighted gross domestic products per capita (P-W GDPs/capita) were calculated by summing the GDPs for all countries that fall into each category (for which GDP data were available), then dividing this number by the sum of the total population for all countries that fall into this category (for which GDP data were available). GDP data for 2020 were drawn from the World Bank [26].

[e] One of the two countries has two regulatory values, one for ordinary circumstances and one for emergencies.

[f] Some countries provide regulations for $O_2$ specified as minimum values, some provide ranges of acceptable values between minimum and maximum values, and some do not specify whether the regulatory values are to be treated as minimum or maximum values. Some other countries regulate $O_2$ in terms of % saturation. Thus, no summary statistics for the regulatory values can be calculated for this substance.

[g] Six countries provide regulations for $P_2O_5$ rather than $PO_4^{3-}$. For the sake of comparison, the $P_2O_5$ regulations were converted to $PO_4^{3-}$ equivalent values.

## Contaminants with World Health Organization Aesthetic Values as well as Health-based Values

The WHO has set both AVs as well as HBVs for certain contaminants that may have noticeable aesthetic effects for consumers. For some of the contaminants that fall into this category, the WHO has also set formal GVs, while for other contaminants, they have declined to set GVs, noting that the aesthetic effects may be detected at levels lower than the HBVs. Contaminants with AVs as well as HBVs are summarized in Table 7. This table also shows whether the WHO has additionally set a formal GV, the number of countries with "mandatory" and "indicator" values for each contaminant, as well as the number of countries that meet or do not meet the WHO's HBV for the contaminant or have no regulation at all.

It should be noted that the WHO AVs are subjective; they are not supported with values drawn from formal laboratory or epidemiological studies of consumer behavior or preferences [18]. In some cases, laboratory studies have reported taste threshold values that are far higher than the WHO AVs, so relying on taste to deter consumption of water whose contamination levels exceed HBVs may not be sufficient to protect public health [342]. For example,

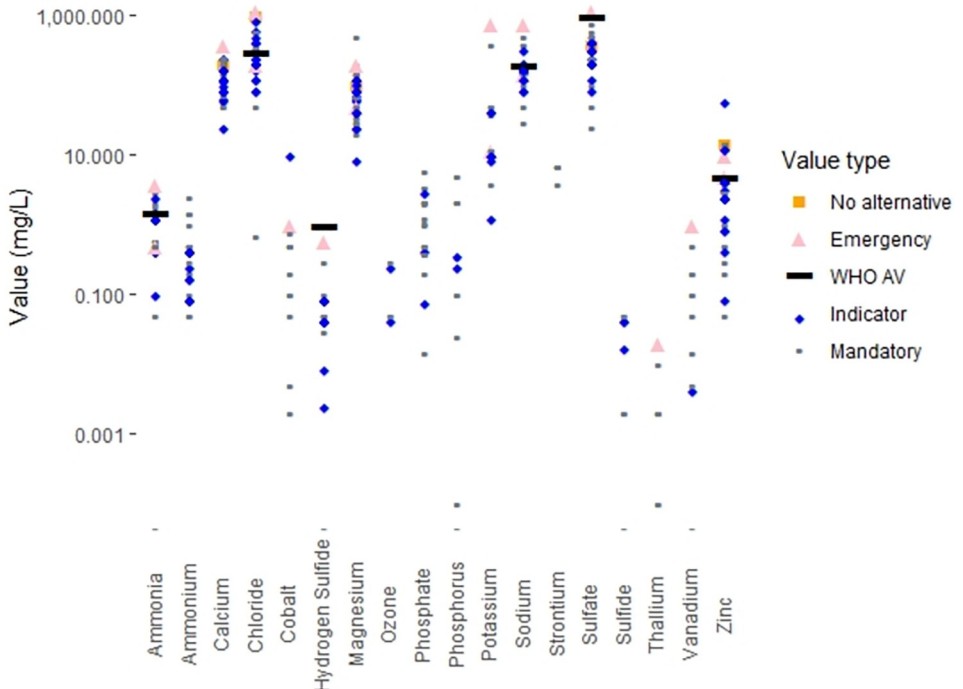

**Fig 15. A visual comparison of the inorganic contaminants that do not have 2022 World Health Organization (WHO) guideline values (GVs), health-based values (HBVs), or health-based reference values (RVs) [15] and appear in five or more national drinking water standards.** Values are shown in milligrams/liter (mg/L) on a logarithmic scale.

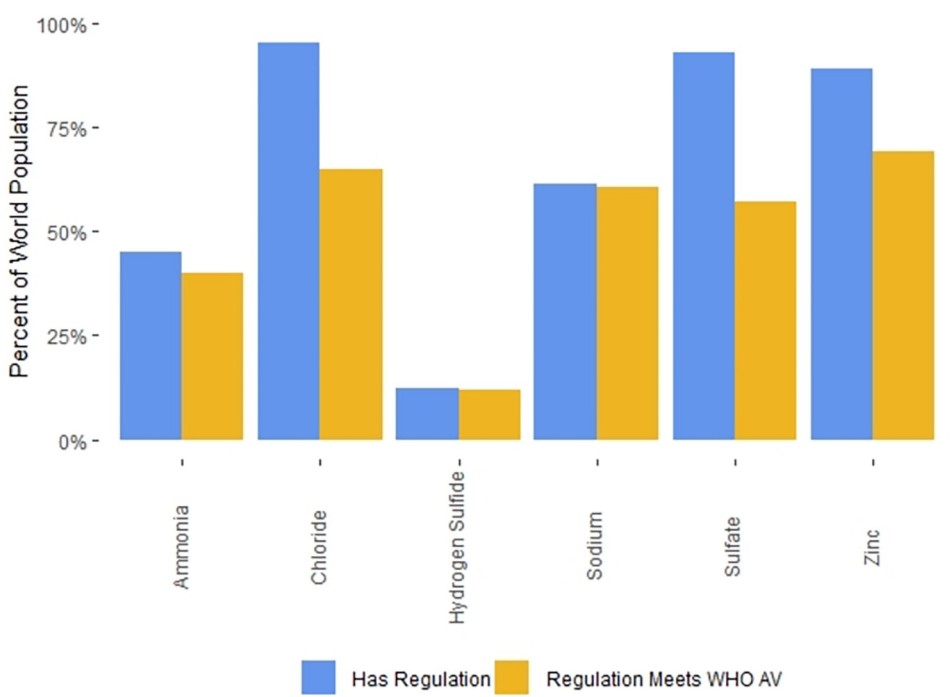

**Fig 16. Percentage of world population living in countries with regulations and whose regulations meet World Health Organization (WHO) aesthetic values (AVs).** Population data are from the World Bank for 2020 [25].

**Table 7. National regulations in milligrams/liter (mg/L) for inorganic contaminants for which the WHO has both Aesthetic Values (AVs) and Health-based Values (HBVs) [18].** Country categories are compared by raw count of countries and then by the percentage of world population within each category. Income differences between categories are compared using population-weighted gross domestic projects (P-W GDP)/capita within each category, shown in United States dollars (USD).

| Contaminant | WHO GV (mg/L) | WHO HBV (mg/L) | WHO AV (mg/L) | Total number of distinct countries with regulations[a] (count; % population[b]; P-W GDP/capita in USD[c]) | Number of countries with Mandatory values (count; % population[b]; P-W GDP/capita in USD[c]) | Number of countries with Indicator values (count; % population[b]; P-W GDP/capita in USD[c]) | Countries with regulations < = WHO HBV (count; % population[b]; P-W GDP/capita in USD[c]) | Countries with regulations > WHO HBV (count; % population[b]; P-W GDP/capita in USD[c]) | Countries with no regulation (count; % population[b]; P-W GDP/capita in USD[c]) |
|---|---|---|---|---|---|---|---|---|---|
| Aluminum (Al) | NA | 0.9 | 0.2 | 146 | 56 | 92 | 145 | 1 | 49 |
| | | | | 94% | 45% | 51% | 93.9% | 0.1% | 6% |
| | | | | $11,238 | $6,183 | $15,563 | $11,230 | $24,812 | $5,361 |
| Copper (Cu) | 2 | 2 | 1–5 | 170 | 147 | 53 | 166 | 4 | 25 |
| | | | | 97.4% | 69% | 39% | 96.7% | 0.7% | 2.6% |
| | | | | $11,123 | $12,414 | $15,285 | $11,150 | $3,988 | $2,012 |
| Iron (Fe) | NA | 2 | 0.3 | 165 | 69 | 100 | 164 | 1 | 30 |
| | | | | 94.6% | 43% | 53% | 94.2% | 0.4% | 5.4% |
| | | | | $11,358 | $6,453 | $15,184 | $11,399 | $1,155 | $2,852 |
| Manganese (Mn) | 0.08 | 0.08 | 0.02 | 163 | 101 | 90 | 70 | 93 | 32 |
| | | | | 97% | 56% | 48% | 18% | 79% | 3% |
| | | | | $11,156 | $6,225 | 16,680 | $34,826 | $5,737 | $2,243 |

Abbreviations: AV = Aesthetic Value; HBV = Health-based Value; mg/L = milligrams/liter; NA = Not available; P-W GDP = population weighted gross domestic product; WHO = World Health Organization.

[a] This column shows the number of distinct countries that have regulations. Some countries have both "mandatory and indicator" regulations for certain contaminants, so the sum of the countries with "mandatory" and "indicator" regulations may exceed the number of distinct countries with regulations.

[b] In this table, "% population" refers to the percentage of total world population that falls into each category. These values were calculated by summing the populations of the individual countries that fall into each category, then dividing by the total population of all 195 countries. Population data for 2020 were drawn from the World Bank [25].

[c] In this table, population-weighted gross domestic products per capita (P-W GDPs/capita) were calculated by summing the GDPs for all countries that fall into each category (for which GDP data were available), then dividing this number by the sum of the total population for all countries that fall into this category (for which GDP data were available). GDP data for 2020 were drawn from the World Bank [26].

consumer taste ratings of water that exceeded the WHO's former 0.4 mg/L HBV for Mn (and far exceeded the current HBV of 0.08 mg/L AV) were not sufficient to cause consumers to reject water in Nepal or Minnesota [343, 344]. Furthermore, concerns about staining of plumbing fixtures mentioned as support for the AVs for Fe are not relevant in regions that lack indoor plumbing fixtures, so they are certainly not sufficient to protect health due to supposed lack of consumer acceptability in these regions [345].

## Conclusions

While existence of regulations does not imply adherence to the regulations, regulations provide a starting point for protecting public health through the establishment of legal criteria that can be used to determine whether water sources are deemed "safe" or "acceptable" [346]. The WHO drinking-water guidelines have an extraordinary influence on international drinking water standards. The WHO drinking water guidelines influence which contaminants are regulated by national standards and the regulatory levels for the contaminants. The WHO guidelines are particularly influential for countries that formally use the guidelines as their national standards. The countries that use the WHO guidelines as their national standards tend to be resource-limited, so the WHO guidelines have an inordinate influence for poor

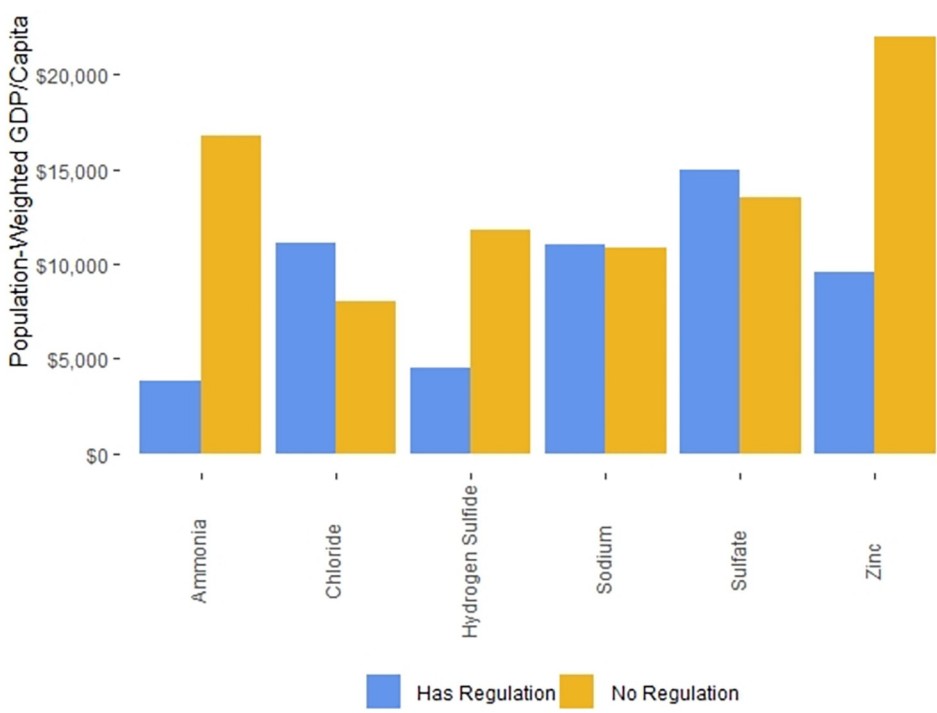

**Fig 17. Population-weighted Gross Domestic Products (P-W GDPs)/capita in United States dollars (USD) for populations living in countries with or without regulations for contaminants for which there are WHO AVs.** Population and GDP data are from the World Bank for 2020 [25, 26].

populations. Because of these strong influences on national drinking water standards, it is critical for the WHO guidelines to accord with the toxicological research and be updated regularly as new research becomes available.

This study examined inorganic contaminants in drinking water. These substances are often found in drinking water due to natural processes, although they may also be sourced anthropogenically. In contrast, pesticides, disinfectants, and their byproducts are only sourced anthropogenically. Of the 34 inorganic drinking water contaminants for which the WHO has noted adverse health effects in its most recent drinking water guidelines publication, only five have values that potentially take into account research published in the last 10 years (Table 1 and Fig 2) [18]. More than half of the point of departure studies used to support the WHO's drinking water guidance values were published in 2010 or earlier (Table 1 and Fig 2). Thus, updated risk assessments are urgently needed for the majority of the inorganic contaminants for which the WHO publishes drinking water guidance.

Most countries (90%) have established drinking water standards, and most of the world's population (98%) live in countries that have drinking water standards. The P-W GDP/capita of the countries without drinking water guidelines is $1,008, highlighting that living in a country without drinking water standards is a plight of the poorest of world citizens. A nation's per capita resources also influences whether the national drinking water standards are freely available to citizens or must be purchased from standards bureaus, and whether the official drinking water standards are likely to be free of typographical errors. Alarmingly, 30% of the world's population can only access their national drinking water standards by purchasing them from their national standards bureau. Furthermore, the burden of having to pay to access one's

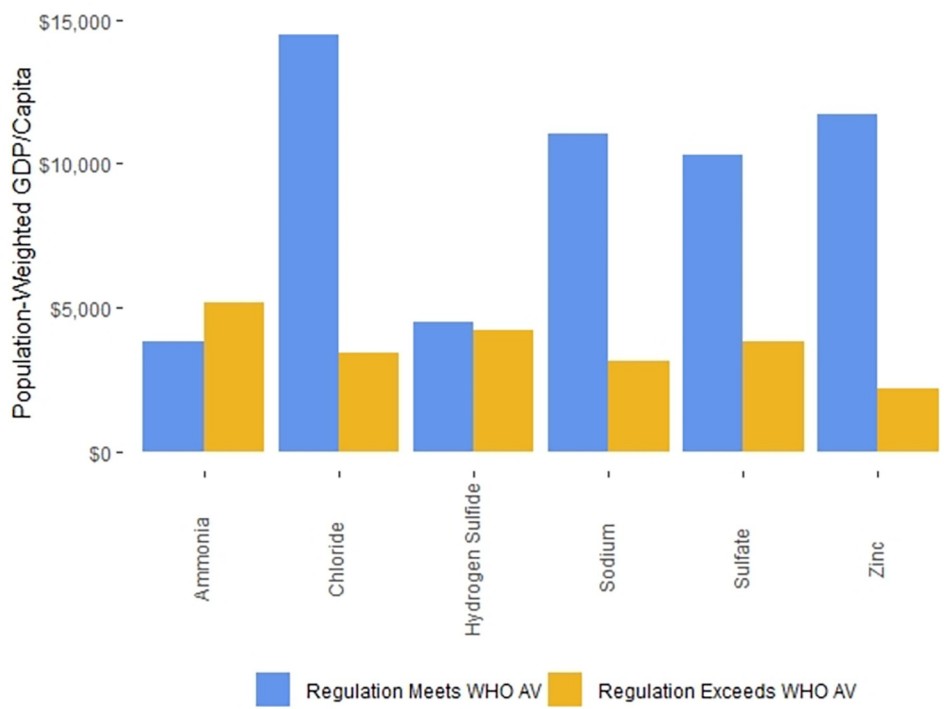

**Fig 18. Population-weighted Gross Domestic Products (P-W GDPs)/capita in United States dollars (USD) for populations living in countries whose regulations for contaminants with World Health Organization (WHO) aesthetic values (AVs) meet or exceed (less protective than) the WHO AVs.** Population and GDP data are from the World Bank for 2020 [25, 26].

national drinking water standards is disproportionately experienced in lower income countries; the P-W GDP/capita for populations living in countries in which national standards can only be accessed through purchase is substantially lower than the P-W GDP/capita for populations living in countries in which the national standards are freely accessible and available for all.

Countries which choose to use the WHO's current drinking water guidelines as their national drinking water standards do not incur the expense of developing and publishing standards, and their citizens have free access to their de facto national drinking water standards through WHO publications. Countries whose national drinking water standards are linked to the WHO's drinking water guidelines also benefit from regular and continuous updates to the WHO's drinking water guidelines, although they do not exercise the option of modifying the guidelines to fit their national situations and priorities, as is inherent in the concept of "guideline".

Countries that develop their own guidelines independently must continuously monitor scientific developments to ensure that their guidelines are up to date with scientific results. Comparison of national drinking water standards to the WHO's GVs can provide an indication of whether the national standards are keeping up with international guidelines. The national standards of the United States stand out as being exceptionally out of date, with the highest number of exceedances (values which are less protective) of WHO GVs of any high-income country (Fig 6).

Of the 16 inorganic substances for which the WHO provides a formal guideline value (GV), all are represented in national drinking water standards, and 15 (94%) are represented in

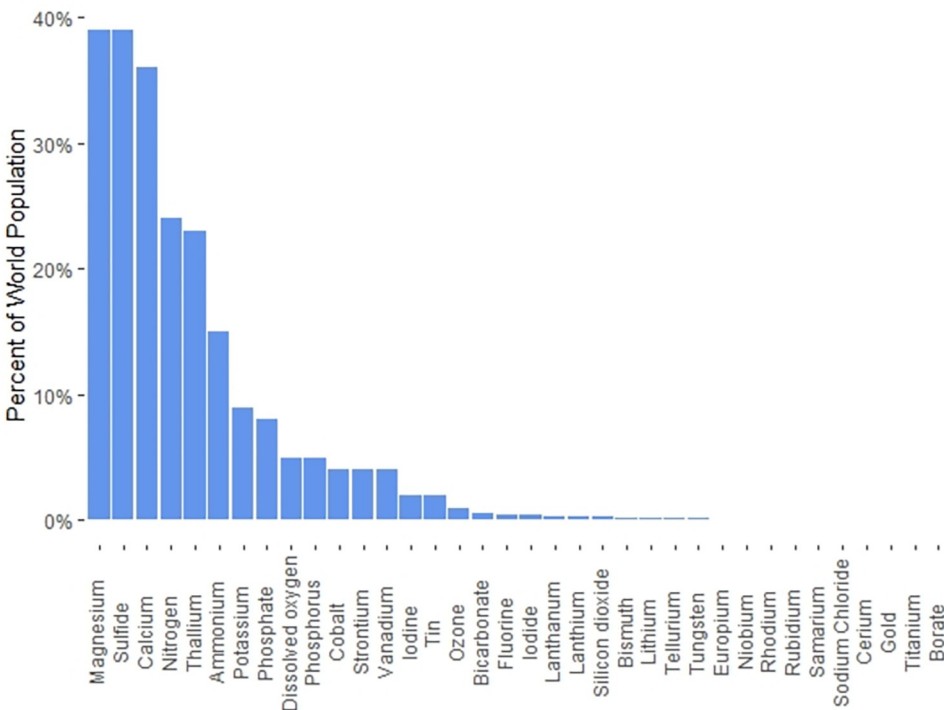

**Fig 19. Percentage of world population living in countries with regulations for inorganic contaminants in drinking water that have no World Health Organization (WHO) guidance.** Population data are from the World Bank for 2020 [25].

national standards that are used by at least 50% of the world's population; see Table 4. Arsenic, Cd, Cr, Cu, $NO_3^-$, and Pb all appear in national regulations that cover more than 95% of the world's population. With the exceptions of Ba, B, Ni, and $NO_3^-$, the P-W GDPs/capita are higher for populations living in countries with regulations for these contaminants than populations living in countries without regulations for these contaminants. Arsenic and Cd stand out as contaminants whose national standards are most closely tied to economic resources, with the P-W GDPs/capita of the populations covered by standards for these contaminants being highest in countries with regulations that meet the WHO's GVs, lower for countries whose regulations are above the WHO's GVs, and lowest for countries without national standards for these contaminants (Table 4 and Fig 9).

All of the six inorganic substances for which the WHO provides health-based values (HBVs) or reference values (RVs) but not formal GVs are also represented in national drinking water standards, with five (Ag, Al, $CN^-$, Fe, and Mo) represented in national standards used by at least 50% of the world's population (Table 5 and Fig 12). The most heavily regulated contaminants with WHO HBVs or RVs but no GVs as measured by the proportion of world population that has national standards are Al, $CN^-$, and Fe. Molybdenum stands out in this group as being a contaminant whose national standards are tied to economic resources, with the P-W GDPs/capita of the population covered by national standards being lowest for countries whose standards meet the WHO's HBV, higher for countries whose standards are above (less protective than) the WHO's HBV, and highest for countries without national standards for Mo (Table 5 and Figs 12 and 13).

There are 43 other inorganic contaminants represented in national drinking water standards that have neither WHO GVs or HBVs, of which six have WHO AVs (Table 6). The most

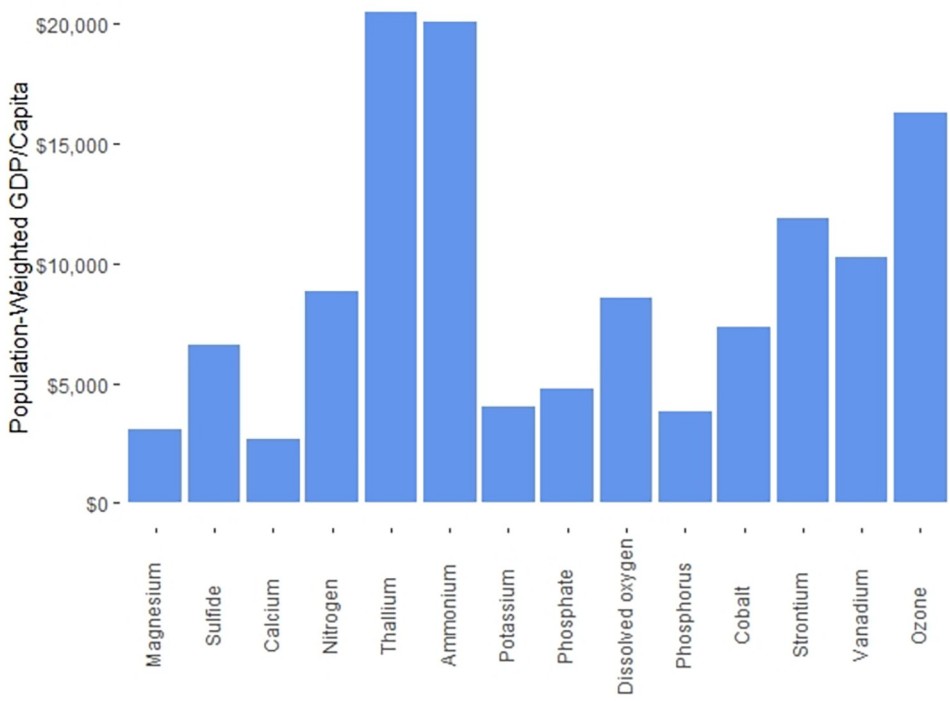

**Fig 20. Population-weighted gross domestic products (P-W GDPs)/capita for populations living in countries with regulations for contaminants for which there are five or more national regulations but no World Health Organization (WHO) guidance.** Population and GDP data are from the World Bank for 2020 [25, 26].

heavily regulated inorganic contaminants with neither WHO GVs or HBVs as measured by the proportion of world population living in countries with national standards are $Cl^-$, Na, $SO_4^{2-}$, and Zn, all of which have WHO AVs (Table 6). Fourteen contaminants without WHO guidance (no GVs, HBVs, or AVs) are represented in the national drinking water standards of five or more countries. Of these 14 contaminants without WHO guidance that are regulated by at least five countries, the P-W GDPs/capita of the populations covered by the national standards are substantially higher for countries that have regulations for Tl, $NH_4^+$, Sr, and $O_3$ than countries that do not (Table 6 and Fig 19).

There are four inorganic contaminants for which the WHO has both HBVs and AVs (Al, Cu, Fe, and Mn); of these two also have GVs (Cu and Mn) and two do not (Al and Fe). Of these four contaminants with both HBVs and AVs, Mn stands out as the contaminant with the highest percentage of world population living in countries that exceed (are less protective than) the HBV (79%) and with the highest skew in P-W GDP/capita for populations living in countries that meet this HBV ($34,826) compared to those living in countries that exceed (are less protective than) the WHO HBV ($2,243).

## Supporting information

**S1 File. Database of international regulations for inorganic chemicals in drinking regulation used in the analyses for this study.**
(XLSX)

**S2 File. R code used to analyze the regulations listed in S1 File.**
(R)

**S3 File. R code used to analyze the countries listed in S1 File.**
(R)

**S4 File. R code used to analyze the contaminants in S1 File.**
(R)

## Acknowledgments

We are grateful to Leif Rasmussen, Esq. for his assistance with international law and to Dr. Mohammad Yusuf Siddiq for help translating Arabic documents.

## Author Contributions

**Conceptualization:** Erika J. Mitchell.

**Data curation:** Erika J. Mitchell, Seth H. Frisbie.

**Formal analysis:** Erika J. Mitchell.

**Funding acquisition:** Seth H. Frisbie.

**Investigation:** Erika J. Mitchell, Seth H. Frisbie.

**Methodology:** Erika J. Mitchell.

**Validation:** Seth H. Frisbie.

**Visualization:** Erika J. Mitchell.

**Writing – original draft:** Erika J. Mitchell.

**Writing – review & editing:** Erika J. Mitchell, Seth H. Frisbie.

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
