## [Decision Letter · Decision Letter 0]

17 May 2023

PONE-D-23-08184A comprehensive survey and analysis of international drinking water regulations for inorganic chemicals with comparisons to the World Health Organization’s drinking-water guidelinesPLOS ONE

Dear Dr. Frisbie,

Thank you for submitting your manuscript to PLOS ONE. After careful consideration, we feel that it has merit but does not fully meet PLOS ONE’s publication criteria as it currently stands. Therefore, we invite you to submit a revised version of the manuscript that addresses the points raised during the review process.

Some of the methodologies need justifications. Please also spell out numbers in the main body of the text if they are less then ten (e.g., nine instead of 9).

We look forward to receiving your revised manuscript.

Kind regards,

Zhi Zhou, Ph.D.

Academic Editor

PLOS ONE

Journal Requirements:

"SHF received support for this project from the Norwich University Board of Fellows Faculty Development Prize. There is no grant number for this support. The Board of Fellows had no role in study design, data collection and analysis, decision to publish, or preparation of the manuscript."

"This study was supported by Norwich University, and Better Life Laboratories, Inc. We are grateful to Leif Rasmussen, Esq. for his assistance with international law and to Dr. Mohammad Yusuf Siddiq for help translating Arabic documents."

"SHF received support for this project from the Norwich University Board of Fellows Faculty Development Prize. There is no grant number for this support. The Board of Fellows had no role in study design, data collection and analysis, decision to publish, or preparation of the manuscript."

"The authors have declared that no competing interests exist. EJM’s affiliation

is with Better Life Laboratories, a nonprofit organization that conducts scientific research and provides technical expertise, equipment, and training to help needy people around the world. Better Life Laboratories received no specific funding for this project from any donors. Donors to Better Life Laboratories provided no input in choosing the subject matter of this project, the hypotheses that were tested, the method of analysis, the research findings, or the manner of disseminating the results."

6. We note that you have referenced (unpublished) on page 13 which has currently not yet been accepted for publication. Please remove this from your References and amend this to state in the body of your manuscript: (ie “Bewick et al. [Unpublished]”) as detailed online in our guide for authors

http://journals.plos.org/plosone/s/submission-guidelines#loc-reference-style.

7. We note that [Figure 3] in your submission contain [map/satellite] images which may be copyrighted. All PLOS content is published under the Creative Commons Attribution License (CC BY 4.0), which means that the manuscript, images, and Supporting Information files will be freely available online, and any third party is permitted to access, download, copy, distribute, and use these materials in any way, even commercially, with proper attribution. For these reasons, we cannot publish previously copyrighted maps or satellite images created using proprietary data, such as Google software (Google Maps, Street View, and Earth). For more information, see our copyright guidelines: http://journals.plos.org/plosone/s/licenses-and-copyright.

a. You may seek permission from the original copyright holder of Figure 3 to publish the content specifically under the CC BY 4.0 license.  

8. Please upload a copy of Supporting Information Figure/Table/etc. Supplementary Table 1 which you refer to in your text on page 15.

Reviewers' comments:

Reviewer's Responses to Questions

**Comments to the Author**

1. Is the manuscript technically sound, and do the data support the conclusions?

Reviewer #1: Partly

2. Has the statistical analysis been performed appropriately and rigorously? 

Reviewer #1: Yes

3. Have the authors made all data underlying the findings in their manuscript fully available?

Reviewer #1: Yes

4. Is the manuscript presented in an intelligible fashion and written in standard English?

Reviewer #1: Yes

5. Review Comments to the Author

Reviewer #1: The manuscript entitles “A comprehensive survey and analysis of international drinking water regulations for inorganic chemicals with comparisons to the World Health Organization’s drinking- water guidelines” needs major revision before publishing in the journal:

- In the page 5, line 141, “μg/L”, and “mg/L” were mentioned as searched terms, are these two terms suitable for finding papers or other publications???!!!

- The type of study was mentioned as research article but the present study is review article. So correct the type of study after that mention the search strategy.

- Mention the full name of elements for the first time after that use abbreviations.

- The aims of the study were not mentioned and the content is not well classified.

- Use the methods of the following studies to improve the manuscript:

- Emerging contaminants migration from pipes used in drinking water distribution systems: a review of the scientific literature. Environmental Science and Pollution Research (2022), pp.1-27.

- Evaluation of occurrence of organic, inorganic, and microbial contaminants in bottled drinking water and comparison with international guidelines: a worldwide review. Environmental Science and Pollution Research (2022), pp.1-15.

6. PLOS authors have the option to publish the peer review history of their article (what does this mean?). If published, this will include your full peer review and any attached files.

Reviewer #1: No

---

## [Author Response · Author response to Decision Letter 0]

9 Jun 2023

PONE-D-23-08184

A comprehensive survey and analysis of international drinking water regulations for inorganic chemicals with comparisons to the World Health Organization’s drinking-water guidelines

PLOS ONE

Dear Dr. Frisbie,

Thank you for submitting your manuscript to PLOS ONE. After careful consideration, we feel that it has merit but does not fully meet PLOS ONE’s publication criteria as it currently stands. Therefore, we invite you to submit a revised version of the manuscript that addresses the points raised during the review process.

Some of the methodologies need justifications. 

***We have addressed this point by answering the specific reviewer comments below.

Please also spell out numbers in the main body of the text if they are less then ten (e.g., nine instead of 9).

***Done.

We look forward to receiving your revised manuscript.

Kind regards,

Zhi Zhou, Ph.D.

Academic Editor

PLOS ONE

Journal Requirements:

***Done

***We have added our R codes to the supplementary materials (supplementary files 2-4). We have added citations to these R codes in the text. The R codes contain comments throughout to assist the readers in understanding our coding.

"SHF received support for this project from the Norwich University Board of Fellows Faculty Development Prize. There is no grant number for this support. The Board of Fellows had no role in study design, data collection and analysis, decision to publish, or preparation of the manuscript."

***As requested by the editor, please change the funding statement to read, “SHF received support for this project from the Norwich University Board of Fellows Faculty Development Prize. There is no grant number for this support. The funders had no role in study design, data collection and analysis, decision to publish, or preparation of the manuscript.”

"This study was supported by Norwich University, and Better Life Laboratories, Inc. We are grateful to Leif Rasmussen, Esq. for his assistance with international law and to Dr. Mohammad Yusuf Siddiq for help translating Arabic documents."

"SHF received support for this project from the Norwich University Board of Fellows Faculty Development Prize. There is no grant number for this support. The Board of Fellows had no role in study design, data collection and analysis, decision to publish, or preparation of the manuscript."

***We have removed the funding-related text from the manuscript. No further changes to the funding statement are necessary since they have been covered in other comments (#3, #5).

"The authors have declared that no competing interests exist. EJM’s affiliation

is with Better Life Laboratories, a nonprofit organization that conducts scientific research and provides technical expertise, equipment, and training to help needy people around the world. Better Life Laboratories received no specific funding for this project from any donors. Donors to Better Life Laboratories provided no input in choosing the subject matter of this project, the hypotheses that were tested, the method of analysis, the research findings, or the manner of disseminating the results."

6. We note that you have referenced (unpublished) on page 13 which has currently not yet been accepted for publication. Please remove this from your References and amend this to state in the body of your manuscript: (ie “Bewick et al. [Unpublished]”) as detailed online in our guide for authors

http://journals.plos.org/plosone/s/submission-guidelines#loc-reference-style.

***We request an exception to this policy concerning the citation of unpublished manuscripts in this case. The only reason why we are making reference to this unavailable document is that the WHO has used it as the basis for its drinking water guideline. The table in which this reference occurs has been constructed as a comprehensive reference list to enable readers to identify all studies on which the WHO guidelines are based; this information has never been collated in published documents before. It was the WHO, not ourselves, that chose to cite an unpublished, unavailable document. In this footnote, we wish to highlight the fact that the guideline is based on an unpublished document, and that we were denied access to this document when we requested it from the agency that created it. For the sake of completeness for the table, it is extremely important that we be able to list the complete reference information as the WHO provided it in their citations documenting their guideline.

7. We note that [Figure 3] in your submission contain [map/satellite] images which may be copyrighted. All PLOS content is published under the Creative Commons Attribution License (CC BY 4.0), which means that the manuscript, images, and Supporting Information files will be freely available online, and any third party is permitted to access, download, copy, distribute, and use these materials in any way, even commercially, with proper attribution. For these reasons, we cannot publish previously copyrighted maps or satellite images created using proprietary data, such as Google software (Google Maps, Street View, and Earth). For more information, see our copyright guidelines: http://journals.plos.org/plosone/s/licenses-and-copyright.

a. You may seek permission from the original copyright holder of Figure 3 to publish the content specifically under the CC BY 4.0 license. 

***We created our map ourselves using the ggmap package in R. As requested in the reference document for the ggmap package (https://cran.r-project.org/web/packages/ggmap/citation.html), we have cited the authors of the ggmap package (Kahle and Wickham, 2013) since they provided the source code for the base maps on a CC by 4.0 basis. The ggmap source code was based on the public domain Natural Earth project. We have added a citation to Kahle and Wickham, 2013 [30] to our figure caption. The map was not made with any copyrighted materials and it is suitable for CC by 4.0 licensing as is.

8. Please upload a copy of Supporting Information Figure/Table/etc. Supplementary Table 1 which you refer to in your text on page 15.

***Done.

Reviewer #1: The manuscript entitles “A comprehensive survey and analysis of international drinking water regulations for inorganic chemicals with comparisons to the World Health Organization’s drinking- water guidelines” needs major revision before publishing in the journal:

- In the page 5, line 141, “μg/L”, and “mg/L” were mentioned as searched terms, are these two terms suitable for finding papers or other publications???!!!

***For many countries, finding copies of the national drinking water regulations was extremely difficult. If an initial search did not turn up a required document, we tried many, many alternative search strategies. One of the most successful strategies was to search “μg/L” or “mg/L” in combination with “drinking water” and the name of the country. This was often a successful search method because many drinking water regulations refer to these units, so the units are contained as text within the document. Since these are international units, they are constant, regardless of the language of the document, so they do not need to be translated into local languages for searching. Note that this search task was used to find national drinking water regulations, which are specific and unique documents. Such a search task is quite different from the open-ended task of identifying research papers on a specific topic.

No changes made.

- The type of study was mentioned as research article but the present study is review article. So correct the type of study after that mention the search strategy.

***Although this project required database searching for published documents, it represents original research, rather than a review article. A review article collates and compares data collected by other scientists; the data have all been previously analyzed before they are reconsidered or summarized in the review. In contrast, the documents that we searched for to create our database were primary regulatory documents. We created a database of datapoints collated from these primary regulatory documents; the datapoints were levels set by regulators and not the results of any prior scientific studies. After we collated the database of regulatory datapoints, we performed original and novel analyses of these datapoints. These analyses revealed new patterns across this entirely novel dataset, patterns which will likely be of interest and use to regulators and drinking water scientists.

No changes made.

- Mention the full name of elements for the first time after that use abbreviations.

***Done

- The aims of the study were not mentioned and the content is not well classified.

***Thank you for pointing out this weakness. We have re-written the final paragraph of our introduction to clarify these points.

- Use the methods of the following studies to improve the manuscript:

- Emerging contaminants migration from pipes used in drinking water distribution systems: a review of the scientific literature. Environmental Science and Pollution Research (2022), pp.1-27.

- Evaluation of occurrence of organic, inorganic, and microbial contaminants in bottled drinking water and comparison with international guidelines: a worldwide review. Environmental Science and Pollution Research (2022), pp.1-15.

***While these are quite interesting and important articles, both of them are scientific reviews. They analyze the results of other scientific studies that were previously published. Our manuscript, in contrast, is not a review, but rather a novel scientific study. Our manuscript analyzes datapoints collated from primary regulatory documents. Although the collation of the regulatory documents involved database and internet searching, the inclusion of a systematic search process in our methods does not make the manuscript a review paper since we did not analyze any previously published research articles. Since the review articles mentioned above were of an entirely different genre, their methods are not relevant for the original research project that we completed.

No changes made.

---

## [Editor Report · Decision Letter 1]

15 Jun 2023

A comprehensive survey and analysis of international drinking water regulations for inorganic chemicals with comparisons to the World Health Organization’s drinking-water guidelines

PONE-D-23-08184R1

Dear Dr. Frisbie,

We’re pleased to inform you that your manuscript has been judged scientifically suitable for publication and will be formally accepted for publication once it meets all outstanding technical requirements.

Kind regards,

Zhi Zhou, Ph.D.

Academic Editor

PLOS ONE
---

## [Editor Report · Acceptance letter]

21 Jun 2023

PONE-D-23-08184R1 

A comprehensive survey and analysis of international drinking water regulations for inorganic chemicals with comparisons to the World Health Organization’s drinking-water guidelines 

Dear Dr. Frisbie:

I'm pleased to inform you that your manuscript has been deemed suitable for publication in PLOS ONE. Congratulations! Your manuscript is now with our production department. 

Kind regards, 

on behalf of

Dr. Zhi Zhou 

Academic Editor

PLOS ONE